# From Mismatch to Harmony: Resolving Feature-Classifier Mismatch in Federated Learning via Prompt-Driven Feature Transformation

## Abstract

In conventional Federated Learning approaches like FedAvg, training a global model becomes challenging in the presence of data heterogeneity. To address this, Personalized Federated Learning (PFL) has emerged as a leading solution, enabling clients to train personalized models that are tailored to local data distributions. Surprisingly, our linear probe experiments reveal that FedAvg's feature extractor outperforms most PFL methods on local client data. Even more intriguingly, applying a simple linear transformation to align local features from FedAvg's extractor with the classifier enables FedAvg to surpass most PFL methods. These findings suggest that in data heterogeneity scenarios, FedAvg's weaker performance is not due to inadequate global model training but rather a mismatch between local features and the classifier. This observation motivates us to develop a new framework to address this mismatch problem. A straightforward solution would be to insert the personalized linear transformation layer mentioned above between the global feature extractor and the global classifier. However, this approach can easily overfit the limited local training data due to the large number of personalized parameters, and it is insufficient for handling complex datasets. In this paper, we introduce FedPFT, which leverages personalized prompts to resolve the mismatch problem. These prompts, along with local features, are fed into a shared self-attention-based module, where features are transformed via the attention mechanism to align with the global classifier. These prompts consist of minimal trainable parameters, reducing the risk of overfitting to local data. Additionally, this prompt-driven approach offers strong flexibility, allowing for task-specific prompts to integrate additional training objectives (e.g., contrastive learning) to further enhance performance. Our experiments demonstrate that FedPFT outperforms state-of-the-art methods by up to 5.07%, with additional improvements of up to 7.08% when collaborative contrastive learning is introduced.

## 1 Introduction

Federated Learning (FL) enables clients to collaboratively train a global model without sharing their raw data. A major challenge in FL is data heterogeneity, where data across clients is not independently and identically distributed (non-IID). This issue results in degraded performance of the global model trained in conventional FL methods such as FedAvg McMahan et al. (2017).

To address this issue, Personalized Federated Learning (PFL) has been proposed, which allows clients to train personalized models to fit their local data distribution better. Many existing PFL methods achieve personalization by personalizing parts of the global model. For example, FedPer Arivazhagan et al. (2019) personalizes classifiers, FedBN Li et al. (2021b) personalizes BN layers, AlignFed Zhu et al. (2024) personalizes feature extractors, and FedCAC Wu et al. (2023) selects parameters susceptible to non-IID effect for personalization.

While these methods show substantial improvements over the global model, an interesting observation emerged from our experiments: **the feature extractor trained by FedAvg outperforms those in**

Table 1: Comparison of different methods. Probe Acc. refers to the accuracy achieved by retraining the classifier with local data. Origin Acc. indicates the accuracy of the original model. Match Acc. represents the accuracy after applying a linear transformation to the features to adapt them to the classifier. All accuracies are obtained on the client testing data. **The disparity between Origin Acc. and Match Acc. indicates the degree of mismatch.** The toy example of the models used to calculate the three types of accuracy are in Appendix A.

| Methods | CIFAR-10, $\alpha = 0.5$ | | | CIFAR-10, $\alpha = 1.0$ | | |
| --- | --- | --- | --- | --- | --- | --- |
| | Probe Acc. | Origin Acc. | Match Acc. | Probe Acc. | Origin Acc. | Match Acc. |
| FedAvg | 72.52% | 59.66% | 72.60% | 68.38% | 60.33% | 68.37% |
| FedPer | 71.07% | 68.86% | 71.03% | 66.51% | 64.83% | 66.75% |
| FedBN | 70.15% | 66.20% | 70.60% | 66.51% | 62.97% | 66.80% |
| FedCAC | 71.56% | 68.71% | 71.63% | 66.98% | 64.90% | 67.11% |
| FedPFT | 72.59% | 72.66% | 72.73% | 69.57% | 69.25% | 69.30% |
| FedPFT+Con | 77.25% | 77.06% | 77.68% | 74.02% | 73.88% | 74.75% |

**most PFL methods on local client data**. Specifically, we conduct linear probe experiments, where each client retrains a linear classifier (probe) behind the FL-trained feature extractor. As evident from Table 1, the Probe Acc. of FedAvg exceeds that of the PFL methods, indicating that the features extracted by FedAvg exhibit superior linear separability. This suggests that FedAvg should have greater potential to outperform PFL methods, contrary to typical expectations.

These findings motivate us to investigate why FedAvg underperforms compared to PFL methods. To unveil this puzzle, we introduce a linear layer between the global feature extractor and global classifier on each client. This layer is retrained with local data to align the features with the classifier. The Match Acc. in Table 1 shows that applying a simple linear transformation to local features significantly improves accuracy over the original model (Origin Acc.). This reveals that there is a mismatch between local features and the global classifier in FedAvg. Interestingly, the Match Acc. of FedAvg even exceeds the Origin Acc. of current PFL methods, indicating that **this mismatch between local features and the classifier is a key reason for FedAvg's suboptimal performance**. A toy example illustrating the mismatch can be seen in Figure 1.

Our experiments with PFL methods in Table 1 (rows 4-6) reveal that these methods indirectly reduce the degree of mismatch, leading to improved Origin Acc. However, there is still a

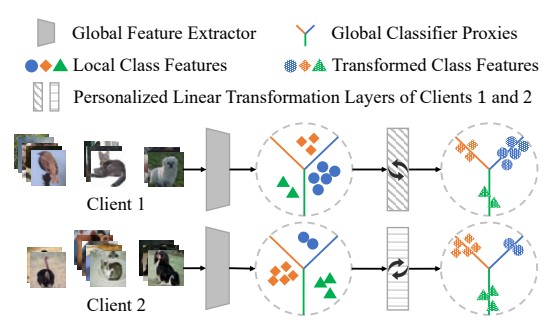

Figure 1: A toy example illustrating the mismatch in FedAvg and how a linear transform addresses it. (a) The local features from two clients have well-clustered structures, but due to the effects of non-IID data, their features mismatch with the global classifier proxies. (b) By applying personalized transformations to the local features of each client, the features are aligned with the global classifier.

notable gap between their Origin Acc. and Match Acc. (up to 4.4%), indicating that the mismatch issue persists. This mismatch not only undermines the model's accuracy during inference but also disrupts the synergy between the feature extractor and classifier during training, ultimately degrading overall performance. These findings suggest that **the mismatch problem is a pervasive yet unresolved challenge in FL**.

To address the mismatch problem during training, we introduce a novel PFL method, FedPFT. While inserting a personalized linear layer between the shared feature extractor and shared classifier is a straightforward approach, it can easily overfit the limited local training data due to the large number of personalized parameters, and it is insufficient for handling complex datasets (as demonstrated in Appendix F). To overcome these limitations, we draw inspiration from prompt technology Jia et al. (2022), which uses prompts to guide model behavior. FedPFT incorporates personalized prompts with minimal trainable parameters and a shared self-attention-based feature transformation module (FTM). Both the prompts and local features are fed into the FTM, where features are transformed via the

attention mechanism. In each round, FedPFT first trains the prompts to align local features with the global classifier. Subsequently, training the model parameters based on this alignment enhances the synergy between the feature extractor and classifier. The results in Table 1 demonstrate that FedPFT not only resolves the mismatch problem but also improves the quality of the feature extractor.

Another advantage of our designed prompt-driven FTM is its strong flexibility across different tasks. It can leverage task-specific prompts to incorporate various tasks beneficial for client collaboration, such as contrastive learning Wang et al. (2023), feature alignment Zhou et al. (2024); Xu et al. (2023), etc. Taking contrastive learning as an example, as shown in Table 1, FedPFT+Con further improves model performance by introducing collaborative contrastive learning through prompts.

Our main contributions can be summarized as follows:

- We identify that the global model's inadequate performance in non-IID scenarios is primarily due to the mismatch between local features and the classifier. We show that the reason personalizing some parameters improves performance is that it indirectly alleviates this issue. This insight offers a new perspective for future PFL approaches to better address the non-IID problem.

- We propose a new PFL framework that incorporates a prompt-driven feature transformation module to align local features with the global classifier. This approach not only resolves the mismatch problem but also provides flexibility for incorporating various collaborative tasks to further enhance PFL performance.

- Our experiments on multiple datasets and non-IID scenarios (including both label shift and feature shift) demonstrate the superiority of FedPFT, outperforming state-of-the-art methods by up to 5.07%. When further incorporating contrastive learning tasks, this improvement can reach up to 7.08%.

## 2 RELATED WORK

PFL is an effective approach to address the challenges of non-IID data in FL. There is a surge of methodologies within PFL, with parameter decoupling methods gaining significant attention due to their simplicity and effectiveness. For a more detailed summary of other categories of PFL methods, please refer to Appendix B.

**Parameter decoupling** methods aim to decouple a subset of parameters from the global model for personalization. Approaches such as FedPer Arivazhagan et al. (2019), FedRep Collins et al. (2021), and GPFL Zhang et al. (2023) focus on personalizing the classifier. In contrast, methods like LG-FedAvg Liang et al. (2020) and AlignFed Zhu et al. (2024) advocate for the personalization of the feature extractor. Additionally, FedBN Li et al. (2021b) and MTFL Mills et al. (2021) propose personalizing batch normalization (BN) layers within the feature extractor. Techniques employing deep reinforcement learning Sun et al. (2021) or hypernetworks Ma et al. (2022) have been used to determine which specific layers to personalize. The recent FedCAC Wu et al. (2023) method advances this by introducing a metric for parameter-wise selection.

These decoupling methods indirectly help alleviate the mismatch issue within the global model by allowing local parameter adjustments. For instance, personalized classifiers involve local adjustments to the classifier to better match the local features extracted by the global feature extractor. However, these methods do not completely resolve the mismatch issue during training. Moreover, personalizing parameters often reduces the extent of client information exchange, which can diminish the overall quality of the feature extractor, thereby limiting the potential benefits of PFL.

## 3 METHODOLOGY

### 3.1 OVERVIEW OF FEDPFT

As illustrated in Figure 2(a), the core of FedPFT is the introduction of a prompt-driven feature transformation module (FTM) $\tau_i$ between the feature extractor $\phi_i$ and the downstream task heads. This module transforms local features during training to match downstream tasks. As shown in Figure 2(b), prompts $p$ and image features $f$ are fed into the FTM, where self-attention operations

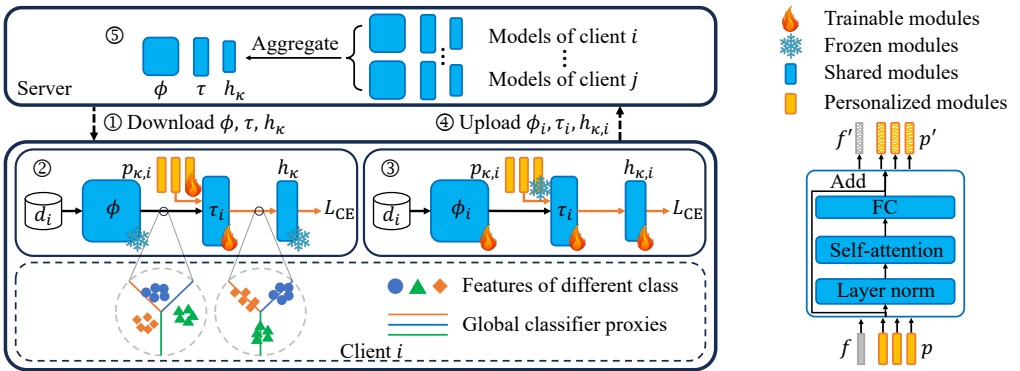

(a) The overview of FedPFT in one communication round     (b) The feature transformation module ($\tau$)

Figure 2: Overview of FedPFT. (a) The training process of each client $i$ in one communication round. (b) The feature transformation module in FedPFT.

transform $f$ to $f'$, which is then used for downstream tasks. FedPFT leverages classification prompts $p_{\kappa,i}$ to transform $f$ to align with the classifier $h_{\kappa,i}$.

Each training round of each client $i$ in FedPFT include five key steps:

1. Client downloads the global models, including feature extractor $\phi$, FTM $\tau$, and classifier $h_\kappa$.
2. Client freezes the global feature extractor $\phi$ and updates $\tau_i$ and prompts $p_{\kappa,i}$ using the cross-entropy loss $L_{\mathrm{CE}}$ to align local features with the frozen global classifier $h_\kappa$.
3. Based on the alignment between the features and the classifier, the client freezes the prompts $p_{\kappa,i}$ and updates $\phi_i, \tau_i, h_{\kappa,i}$ with $L_{\mathrm{CE}}$ to learn client local knowledge.
4. Client uploads $\{\phi_i, \tau_i, h_{\kappa,i}\}$ to the server while retaining $\{p_{\kappa,i}\}$ locally.
5. The server aggregates the models uploaded by the clients.

### 3.2 PROBLEM FORMULATION

In PFL, $N$ clients train their personalized models $w_i, i \in [N]$ under the coordination of a server, aiming for each $w_i$ to perform well on client data distribution $\mathbb{D}_i$. This objective can be formalized as $\min_{\{w_i\}_{i \in [N]}} \frac{1}{N} \sum_{i=1}^{N} L_i(w_i; \mathbb{D}_i)$, where $L_i$ represents the loss function of the $i$-th client.

In this paper, our goal is to enhance personalized models by addressing the mismatch problem between local features and the classifier in the global model. Thus, the training objective of FedPFT can be formulated as:

$$\min_{\phi, \tau, h_\kappa} \min_{\{p_{\kappa,i}\}_{i \in [N]}} \mathbf{E}_i \{L_i(\phi, \tau, h_\kappa, p_{\kappa,i}; d_i) := \mathbf{E}_{d_i}[L_{\mathrm{CE}}(\phi, \tau, h_\kappa, p_{\kappa,i}; d_i)]\}, \tag{1}$$

where $\phi$ and $h_\kappa$ represent the global feature extractor and global classifier, respectively. $\tau$ is the newly introduced global feature transformation module. This module, along with the personalized classification prompt $p_{\kappa,i}$, transforms local features to align with the global classifier. $L_{\mathrm{CE}}$ denotes the cross-entropy loss for classification tasks. $d_i$ represents the local data of the client.

### 3.3 FEATURE TRANSFORMATION MODULE

In FedPFT, we introduce a global feature transformation module (FTM) $\tau$, along with a set of personalized prompts $p_{\kappa,i}$ for each client $i$, to align the features extracted by the global feature extractor $\phi$ with the global classifier $h_\kappa$.

Formally, given a sample $x_j \in d_i$, extracted by the feature extractor $\phi$, the obtained feature is $f_j \in \mathbb{R}^m$, where $m$ is the feature dimension. A collection of $n$ prompts is denoted as $p = \{\boldsymbol{p}^k \in \mathbb{R}^m | k \in \mathbb{N}, 1 \le k \le n\}$. The operation of the FTM is formulated as

$$[f_j', p'] = \tau([f_j, p]), \tag{2}$$

where $[\cdot, \cdot]$ signifies stacking and concatenation along the sequence length dimension, yielding $[f'_j, p'] \in \mathbb{R}^{(1+n) \times m}$. The $f'_j$ represents the transformed feature. An example of the FTM is illustrated in Figure 2(b). In FedPFT, we denote $n_\kappa$ as the number of prompts contained in $p_{\kappa,i}$.

The FTM essentially customizes local features for downstream tasks, offering strong flexibility. It can introduce tasks beneficial for client collaboration by employing various task-specific prompts $p$. We illustrate this in the Section 3.6.

### 3.4 CLASSIFICATION TASK WITH PERSONALIZED PROMPTS

To address the mismatch between local features and the global classifier, FedPFT employs a set of personalized prompts $p_{\kappa,i}$ as inputs to the FTM, transforming each client $i$'s local features to align with the global classifier. Specifically, the classification loss in each client $i$ is defined as:

$$L_{\text{CE}}(\phi, \tau, p_{\kappa,i}, h_\kappa; x, y) = -\log \sum_{c=1}^{C} y_c \log(o_{i,c}), \text{where } x, y \sim d_i. \tag{3}$$

$C$ is the number of classes, and $o_i = \text{Softmax}(h_\kappa \circ \tau([\phi(x), p_{\kappa,i}]))$ represents the predicted probabilities, with $o_{i,c}$ being the ones of class $c$. Details on coordinating the training of the model and prompts to achieve feature and classifier alignment are discussed in Section 3.5.

### 3.5 ALTERNATING TRAINING STRATEGY

To effectively resolve the mismatch problem and coordinate the training of different modules in FedPFT, we propose an alternating training strategy, which partitions each local training round into two phases: the feature transformation phase and the model training phase.

**Feature transformation phase.** In this phase, the training objective is:

$$\min_{p_{\kappa,i}, \tau_i} L_{\text{CE}}(\tau_i, p_{\kappa,i}; \phi_i, h_\kappa, d_i), \tag{4}$$

which aims at training the classification prompts $p_{\kappa,i}$ and $\tau_i$ with the frozen global feature extractor $\phi$ and global classifier $h_\kappa$ to align local features with the classifier.

**Model training phase.** Following the above phase, the goal of this phase is to train the model parameters based on the aligned features and classifier to learn the local knowledge of each client. The training objective is

$$\min_{\phi_i, \tau_i, h_{\kappa,i}} L_{\text{CE}}(\phi_i, \tau_i, h_{\kappa,i}; p_{\kappa,i}, d_i). \tag{5}$$

Let $R$ represent the total number of local epochs in one training round. We divide it into $R_f$ epochs for the feature transformation phase and $R_a$ epochs for the model training phase, where $R_f + R_a = R$. It is crucial that $R_f$ is always larger than $R_a$ to ensure that the mismatch between local features and the classifier is resolved before training the model parameters.

Upon completing local training, the parameters $\phi_i$, $\tau_i$, and $h_{\kappa,i}$ are aggregated at the server to facilitate client collaboration, while $p_{\kappa,i}$ remains locally. We simply adopt the aggregation method used in FedAvg. The pseudo-code of FedPFT is summarized in Algorithm 1.

### 3.6 FEDPFT WITH ADDITIONAL TASKS: AN EXAMPLE OF CONTRASTIVE LEARNING

As discussed in Section 3.3, our FTM provides strong flexibility. By inputting different prompts, it can transform features to adapt to various downstream tasks. Benefiting from this, the FedPFT framework also offers great flexibility, as it can easily incorporate tasks beneficial for PFL, such as contrastive learning Wang et al. (2023), feature alignment Zhou et al. (2024); Xu et al. (2023), multi-task learning, etc., by simply using task-specific prompts. Each task uses its own prompt to transform features, effectively reducing interference between tasks. In this section, we use contrastive learning as an example to illustrate how our method integrates with other tasks.

As depicted in Figure 3, we introduce another set of personalized prompts $p_{\rho,i}$, which are fed into $\tau_i$ to transform features for the contrastive learning task with a global projection head $h_{\rho,i}$. The goal is

---

**Algorithm 1** FedPFT

---

**Input:** Each client's initial personalized prompts $p_{\kappa,i}^{(0)}$; The initial global models $\{\phi^{(0)}, \tau^{(0)}, h_\kappa^{(0)}\}$; Client Number $N$; Total round $T$; Epochs of two learning phases $R_f$ and $R_a$.

**Output:** Personalized model $\{\phi^{(T)}, \tau^{(T)}, h_\kappa^{(T)}, p_{\kappa,i}^{(T)}\}$ for each client.

**for** $t = 0$ to $T - 1$ **do**

    **Client-side:**

    **for** $i = 1$ to $N$ **in parallel do**

        Initializing $\{\phi_i^{(t)}, \tau_i^{(t)}, h_{\kappa,i}^{(t)}\}$ with $\{\phi^{(t)}, \tau^{(t)}, h_\kappa^{(t)}\}$.

        Updating $\{\tau_i^{(t)}, p_{\kappa,i}^{(t)}\}$ by Eq.(4) for $R_f$ epochs to obtain $\{\tau_i^{(t')}, p_{\kappa,i}^{(t+1)}\}$.

        Updating $\{\phi_i^{(t)}, \tau_i^{(t')}, h_{\kappa,i}^{(t)}\}$ by Eq.(5) for $R_a$ epochs to obtain $\{\phi_i^{(t+1)}, \tau_i^{(t+1)}, h_{\kappa,i}^{(t+1)}\}$.

        Sending $\{\phi_i^{(t+1)}, \tau_i^{(t+1)}, h_{\kappa,i}^{(t+1)}\}$ to the server.

    **end for**

    **Server-side:**

    Aggregating a set of global model $\{\phi^{(t+1)}, \tau^{(t+1)}, h_\kappa^{(t+1)}\}$.

    Sending $\{\phi^{(t+1)}, \tau^{(t+1)}, h_\kappa^{(t+1)}\}$ to each client $i$.

**end for**

---

for all clients to collaborate in optimizing the contrastive learning task, improving the performance of the feature extractor. The training objective can be formulated as

$$\min_{\phi,\tau,h_\kappa} \min_{\{p_{\kappa,i}\}_{i\in[N]}} \mathbf{E}_i\{L_i(\phi,\tau,h_\kappa,p_{\kappa,i};d_i) := \mathbf{E}_{d_i}[L_{\text{CE}}(\phi,\tau,h_\kappa,p_{\kappa,i};d_i) + L_{\text{Con}}(\phi,\tau;d_i)]\}, \quad (6)$$

where $L_{\text{Con}}$ represents the contrastive learning loss function. The optimization process for this objective and the definition of $L_{\text{Con}}$ can be found in Appendix C.

As shown in Figure 3 and Eq.(6), FedPFT can easily incorporate other tasks beneficial to PFL by using task-specific prompts. In Section 4, we refer to the above method as FedPFT+Con to validate the effectiveness of combining FedPFT with contrastive learning tasks. In Section G of the Appendix, we also present experimental evidence showing that our FTM effectively coordinates the training of both tasks, enabling them to mutually enhance each other.

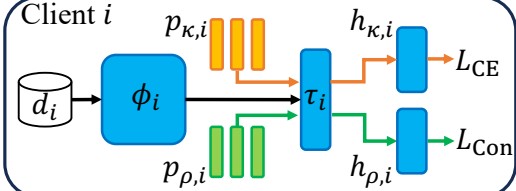

Figure 3: FedPFT with contrastive learning. The orange solid line and the green solid line represent the forward propagation of the classification task and the contrastive learning task, respectively.

## 4 EXPERIMENTS

### 4.1 EXPERIMENTAL SETUP

**Datasets.** In this section, we mainly verify FedPFT in the label shift non-IID scenario, which is one of the most commonly used scenarios in FL research. Specifically, we examine two settings: Dirichlet non-IID and Pathological non-IID. In each setting, we employ three datasets: CIFAR-10 Krizhevsky et al. (2010), CIFAR-100 Krizhevsky et al. (2009), and Tiny ImageNet Le & Yang (2015). We also verify FedPFT in the feature shift non-IID scenario in Appendix I.

In our experiments, each client is assigned 500 training samples. For CIFAR-10 and CIFAR-100 datasets, each client has 100 test samples; for the Tiny ImageNet dataset, each client has 200 test samples. Both training and test data have the same label distribution.

**Baseline methods.** We compare our method against nine state-of-the-art (SOTA) methods: FedAMP Huang et al. (2021), FedPer Arivazhagan et al. (2019), FedRep Collins et al. (2021), FedBN Li et al. (2021b), FedRoD Chen & Chao (2022), pFedSD Jin et al. (2022), pFedGate Chen et al. (2023), FedCAC Wu et al. (2023), and pFedPT Li et al. (2023a). These methods cover the advancements in mainstream PFL research directions.

**Hyperparameter settings.** For the general hyperparameters of FL, we set the number of clients $N = 40$, batch size $B = 100$, and local update rounds $R = 5$. In all datasets, we fix the total rounds $T = 1000$ for each experiment to ensure all methods reach full convergence. The experimental result

is determined by selecting the highest average accuracy achieved by all clients across all rounds. Each experiment is repeated with three random seeds, and the mean and standard deviation are reported. We employ the ResNet He et al. (2016) model architecture, specifically ResNet-8 for CIFAR-10 and ResNet-10 for CIFAR-100 and Tiny ImageNet.

For more details on the experimental setup, please refer to Appendix D.

## 4.2 Comparison with State-of-the-art Methods

We compare our proposed FedPFT with two baseline methods and nine SOTA methods across three datasets and two non-IID scenarios. The experimental results on CIFAR-100 and Tiny ImageNet in Dirichlet non-IID scenario are presented in Table 2. More results on the CIFAR-10 dataset and in Pathological non-IID scenarios are presented in Appendix E.

Table 2: Test accuracy (%) of different methods under Dirichlet non-IID on CIFAR-100 and Tiny ImageNet.

| | CIFAR-100 | | | Tiny ImageNet | | |
|---|---|---|---|---|---|---|
| Methods | $\alpha = 0.1$ | $\alpha = 0.5$ | $\alpha = 1.0$ | $\alpha = 0.1$ | $\alpha = 0.5$ | $\alpha = 1.0$ |
| FedAvg | 34.91±0.86 | 32.78±0.23 | 33.94±0.39 | 21.26±1.28 | 20.32±0.91 | 17.20±0.54 |
| Local | 47.61±0.96 | 22.65±0.51 | 18.76±0.63 | 24.07±0.62 | 8.75±0.30 | 6.87±0.28 |
| FedAMP | 46.68±1.06 | 24.74±0.58 | 18.22±0.41 | 27.85±0.71 | 10.70±0.32 | 7.13±0.21 |
| FedPer | 51.38±0.94 | 28.25±1.03 | 21.53±0.50 | 32.33±0.31 | 12.69±0.42 | 8.67±0.40 |
| FedRep | 51.25±1.37 | 26.97±0.33 | 20.63±0.42 | 30.83±1.05 | 12.14±0.28 | 8.37±0.25 |
| FedBN | 54.35±0.63 | 36.94±0.94 | 33.67±0.12 | 33.34±0.71 | 19.61±0.35 | 16.57±0.44 |
| FedRoD | 60.17±0.48 | 39.88±1.18 | 36.80±0.56 | 41.06±0.77 | 25.63±1.11 | 22.32±1.13 |
| pFedSD | 54.14±0.77 | 41.06±0.83 | 38.27±0.20 | 39.31±0.19 | 19.25±1.80 | 15.91±0.33 |
| pFedGate | 48.54±0.39 | 27.47±0.79 | 22.98±0.03 | 37.59±0.39 | 24.09±0.67 | 19.69±0.14 |
| FedCAC | 57.22±1.52 | 38.64±0.63 | 32.59±0.32 | 40.19±1.20 | 23.70±0.28 | 18.58±0.62 |
| pFedPT | 43.21±1.66 | 35.23±0.87 | 36.25±0.37 | 23.55±0.68 | 22.35±0.49 | 21.69±0.24 |
| FedPFT | **60.98±0.39** | **44.87±0.76** | **41.83±0.37** | **41.49±0.10** | **28.61±0.40** | **25.10±0.59** |

**Results in Dirichlet non-IID scenario.** In this setting, by varying $\alpha$, we can evaluate the performance of methods under different non-IID degrees. The results, as detailed in Table 2, demonstrate that performance varies significantly depending on the underlying design principles of each method. Among all methods, FedRoD demonstrates robust performance across all datasets and non-IID degrees. This is attributed to its design of two classifiers: a personalized classifier for local feature alignment and a global classifier for assistance from other clients to improve generalization. *FedPFT addresses the mismatch issue specifically and achieves superior results across all scenarios.*

We also include FedPFT+Con in the experiments to validate the benefits of combining FedPFT with other tasks. As shown in Table 3, FedPFT+Con further improves performance by incorporating contrastive learning into FedPFT, *significantly outperforming SOTA methods by up to 7.08%.* This improvement is attributed to the strong flexibility of our proposed FTM.

Table 3: Test accuracy (%) of FedPFT+Con under Dirichlet non-IID on CIFAR-100 and Tiny ImageNet.

| | FedPFT+Con | | |
|---|---|---|---|
| Datasets | $\alpha = 0.1$ | $\alpha = 0.5$ | $\alpha = 1.0$ |
| CIFAR-100 | 62.03±1.41 | 47.98±0.78 | 44.29±0.74 |
| Tiny | 43.42±1.62 | 32.44±0.58 | 27.84±0.41 |

## 4.3 Ablation Study

In this section, we validate the effectiveness of each component of our method on the CIFAR-100 dataset under two non-IID degrees. The experimental results are illustrated in Table 4.

Setting I represents FedAvg. Setting II incorporates classification prompts $p_\kappa$ to allow each client to adjust the global model individually to obtain a personalized model, resulting in a performance improvement. Setting III (i.e., FedPFT) incorporates alternating training, where prompts are firstly updated to align local features with the global classifier to address the mismatch problem, followed by training model parameters. This approach effectively resolves the mismatch issue and enhances the

Table 4: Experiments on the CIFAR-100 to illustrate the effectiveness of different modules.

| | $\alpha = 0.1$ | | | | | $\alpha = 0.5$ | | | | |
|---|---|---|---|---|---|---|---|---|---|---|
| Settings | $p_\kappa$ | Alter. | $L_{\text{Con}}$ | $p_\rho$ | Accuracy (%) | $p_\kappa$ | Alter. | $L_{\text{Con}}$ | $p_\rho$ | Accuracy (%) |
| I | | | | | 33.87±1.35 | | | | | 30.09±0.31 |
| II | ✓ | | | | 40.97±1.28 | ✓ | | | | 31.45±1.35 |
| III (FedPFT) | ✓ | ✓ | | | 60.98±0.39 | ✓ | ✓ | | | 44.87±0.76 |
| IV | ✓ | ✓ | ✓ | | 61.13±0.50 | ✓ | ✓ | ✓ | | 47.67±1.42 |
| V (FedPFT+Con) | ✓ | ✓ | ✓ | ✓ | 62.03±1.41 | ✓ | ✓ | ✓ | ✓ | 47.98±0.78 |
| VI | ✓ | | ✓ | ✓ | 53.76±0.35 | ✓ | | ✓ | ✓ | 39.29±1.00 |

synergy between the feature extractor and classifier, *resulting in the greatest performance improvement (up to 20.01%) compared to other modules*.

Setting IV adds contrastive learning loss to Setting III, primarily focusing on enhancing the feature extractor's performance through contrastive learning techniques. Setting V (i.e., FedPFT+Con) incorporates specific prompts $p_\rho$ to better transform features for the contrastive learning task, reducing mutual interference between the two tasks during training. This approach is particularly effective in scenarios with strong non-IID data (e.g., $\alpha = 0.1$).

Setting VI represents incorporating contrastive learning into PFL without addressing the mismatch issue. As shown, while contrastive learning can improve model accuracy by enhancing the quality of the feature extractor, its performance is far inferior to FedPFT (i.e., Setting III). This further *highlights the importance of addressing the mismatch problem in the PFL*.

## 4.4 LEARNED FEATURES OF DIFFERENT METHODS

In this section, we visually compare the quality of features extracted by different methods and highlight the impact of different modules in FedPFT on feature extraction. We conduct experiments on the CIFAR-10 dataset with 10 clients, each allocated 1000 training images and 500 testing images. The data distribution is shown in Figure 4(a). For each method, we visualize the feature vectors of testing data from different clients using t-SNE Van der Maaten & Hinton (2008). The visualization results are shown in Figure 4(b)-(h). In these figures, different colors indicate various data categories, while distinct markers represent different clients, as explained in Figure 4(a).

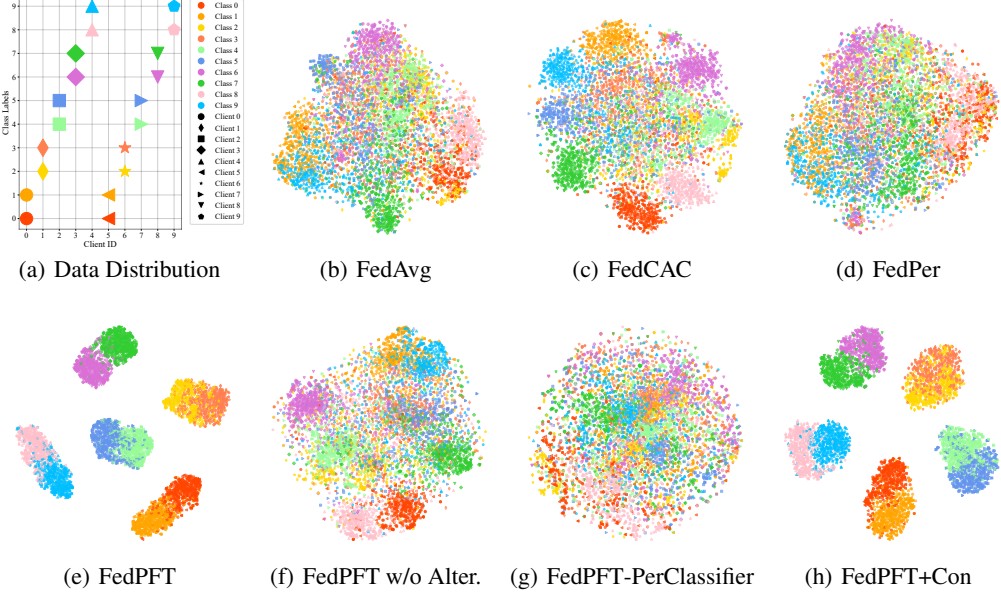

(a) Data Distribution     (b) FedAvg     (c) FedCAC     (d) FedPer

(e) FedPFT     (f) FedPFT w/o Alter.     (g) FedPFT-PerClassifier     (h) FedPFT+Con

Figure 4: t-SNE visualization of features extracted by different methods on the CIFAR-10 dataset.

FedAvg and FedCAC exhibit noticeable cluster structures of features but do not have clear discriminative boundaries. FedPer, on the other hand, shows overlapping features across various classes due

to the use of personalized classifiers that generate unique local feature spaces for each client. As a result, data from different classes across different clients are mapped to similar positions. This interference among clients reduces the effectiveness of the global feature extractor.

FedPFT shows clearer discriminative boundaries, which is attributed to the alignment of local features with the global classifier achieved during local training. We also observe that data from the same class across different clients are mapped to the same positions in the feature space, indicating that the global classifier provides a unified feature space for all clients. Adapting local features to this space can align the training objectives among clients, promoting collaboration among clients.

'FedPFT w/o Alter.' represents not using alternating training. While it shows better clustering than FedAvg, the discriminative quality of the boundaries is weaker compared to FedPFT. This configuration shows increased interference among client models, as it does not fully address the mismatch problem. 'FedPFT-PerClassifier' refers to the use of personalized classifiers. In this case, the feature space becomes highly scattered. This is because the prompt $p_\kappa$ is trained to adapt to personalized classifiers first, it amplifies the variability in feature spaces across clients. FedPFT+Con further introduces contrastive learning into FedPFT, which enhances feature separability.

## 4.5 FEATURE SEPARABILITY OF DIFFERENT METHODS

In this section, we delve deeper into the linear separability of features extracted by various PFL methods. Linear separability is a critical measure of feature quality, indicating the ability of a model to distinguish between classes using simple linear classifiers. We conduct linear probing experiments on the CIFAR-10 and CIFAR-100 datasets to assess this metric, with results detailed in Table 5.

Table 5: Linear probe accuracy (%) of different methods.

| | CIFAR-10 | | | CIFAR-100 | | |
|---|---|---|---|---|---|---|
| Methods | $\alpha = 0.1$ | $\alpha = 0.5$ | $\alpha = 1.0$ | $\alpha = 0.1$ | $\alpha = 0.5$ | $\alpha = 1.0$ |
| FedAvg | 85.01% | 72.52% | 68.38% | 59.50% | 37.40% | 32.33% |
| FedPer | 84.44% | 71.07% | 66.51% | 52.09% | 26.61% | 20.51% |
| FedBN | 84.52% | 70.15% | 66.51% | 57.86% | 35.24% | 30.28% |
| FedCAC | 85.22% | 71.56% | 66.98% | 56.86% | 34.64% | 29.35% |
| FedRoD | 82.79% | 67.07% | 63.12% | 56.88% | 33.99% | 29.22% |
| pFedSD | **85.86**% | 72.42% | 68.12% | 60.07% | 37.33% | 31.99% |
| FedPFT | 85.52% | **72.59**% | **69.57**% | **61.60**% | **43.14**% | **38.47**% |

It can be observed that the feature linear separability of most PFL methods is inferior to FedAvg. This indicates that although they partially alleviate the mismatch issue and achieve better model performance, the quality of the feature extractor is inevitably compromised due to their design, constraining the full potential of PFL.

In stark contrast, FedPFT significantly improves the linear separability of features compared to FedAvg. It accomplishes this by fundamentally addressing the mismatch issue during the training process rather than merely adapting the model post hoc. This proactive approach ensures that the feature extractor not only aligns with the global classifier but also enhances the synergy between them during training, thereby preserving its ability to generalize across diverse data distributions. Consequently, FedPFT enhances both the performance and the utility of the feature extractor.

## 4.6 EFFECT OF PROMPTS

As discussed in the previous sections, two factors influence model performance during the inference phase: 1) the alignment of local features with downstream tasks (e.g., the classifier) and 2) the separability of these features. In this section, we delve into how prompts enhance model performance during the inference phase.

**The impact on the alignment of features with downstream tasks.** We visualize the features transformed by different prompts in FedPFT+Con using t-SNE. The experimental setup is consistent with Section 4.4. The results are depicted in Figure 5. Larger markers in the figures represent feature centroids of corresponding classes for each client.

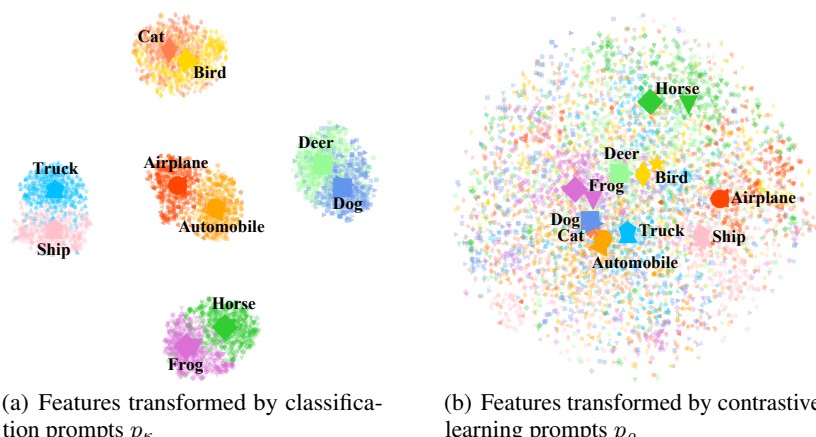

(a) Features transformed by classification prompts $p_\kappa$

(b) Features transformed by contrastive learning prompts $p_\rho$

Figure 5: The effect of different prompts on feature space.

It is evident that the characteristics of features transformed by different prompts are closely related to the downstream tasks. Features obtained from classification prompts $p_\kappa$ are not significantly correlated with image similarity but rather with the distribution of client data. For example, two classes within a client may be close together, regardless of whether the images themselves are truly similar. Additionally, there are clear decision boundaries between features of different classes, which is consistent with the nature of classification tasks. Conversely, features transformed by contrastive learning prompts $p_\rho$ are more related to image similarity. For instance, in Figure 5(b), the feature centroids of 'cat' and 'dog' are close, while those of 'airplane' and 'dog' are far apart, which aligns with the principles of contrastive learning.

Table 6: The effect of prompts $p_\kappa$ and $p_\rho$ on linear probe accuracy (%).

| | CIFAR-10 | | | CIFAR-100 | | |
|---|---|---|---|---|---|---|
| Prompt Type | $\alpha = 0.1$ | $\alpha = 0.5$ | $\alpha = 1.0$ | $\alpha = 0.1$ | $\alpha = 0.5$ | $\alpha = 0.5$ |
| None | 87.69% | 77.12% | 73.93% | 64.08% | 46.50% | 40.79% |
| $p_\kappa$ | 87.83% | 77.25% | 74.02% | 64.12% | 46.43% | 40.95% |
| $p_\rho$ | 87.82% | 77.25% | 74.02% | 64.18% | 46.40% | 40.95% |

**The impact on the linear separability of features.** We conduct linear probe experiments using the CIFAR-10 and CIFAR-100 datasets. The results are detailed in Table 6. We calculate the linear separability of features at three points during the forward propagation process: 'None' represents the features output by $\phi$. '$p_\kappa$' represents the features transformed by the FTM using the classification prompt. '$p_{\rho,i}$' represents the features transformed by the FTM using the contrastive learning prompts. Interestingly, the accuracies across different prompt conditions are generally similar, suggesting that the use of either type of prompt does not significantly impact the feature separability.

The above experiments illustrate that during the inference phase, *prompts work by transforming features into the required format to align with downstream tasks*, rather than improving feature separability. This finding is consistent with the motivation of our paper and highlights the flexibility and adaptability of our designed FTM. It can integrate various client-collaborative tasks, which is beneficial for enhancing the performance of personalized models through task-specific prompts.

## 5 CONCLUSION AND DISCUSSION

We observe that the feature extractor from FedAvg surpasses those in most PFL methods, yet it suffers from inadequate performance due to a mismatch between the local features and the classifier. This mismatch issue not only impacts the performance during model inference but also affects the synergy between the feature extractor and the classifier during training. We propose a new PFL method called FedPFT with a prompt-driven feature transform module to address these issues during training. Our experiments demonstrate that FedPFT not only resolves the mismatch issue but also significantly improves the quality of the feature extractor, achieving substantial performance gains compared to state-of-the-art methods. We discuss the limitations and our future work in Appendix O.

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

## A    MORE EXPLANATION OF THE EXPERIMENTS DISCUSSED IN THE INTRODUCTION

To facilitate understanding of the experiments mentioned in the introduction, we give a toy example to visualize the models used to calculate Origin Acc., Probe Acc., and Match Acc. in Figure 6.

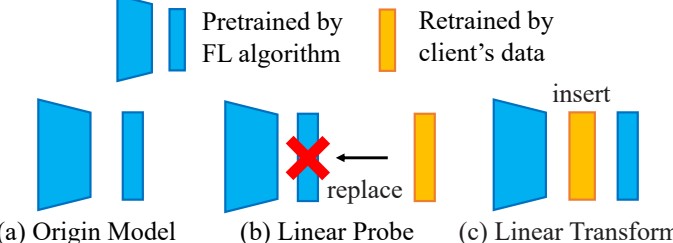

(a) Origin Model    (b) Linear Probe    (c) Linear Transform

Figure 6: A toy example illustrating the model structures used to calculate Origin Acc., Probe Acc., and Match Acc.

Figure 6(a) represents the model trained using the FL algorithm, where the accuracy measured on the client's local data is referred to as Origin Acc. Figure 6(b) illustrates the model obtained from the linear probe experiment, where the classifier in the FL pre-trained model is replaced by a linear classifier retrained on the client data. The accuracy corresponding to this model is referred to as Probe Acc. Figure 6(c) depicts the model obtained from the linear transform experiment, where a linear transformation layer, retrained on client data, is inserted between the feature extractor and classifier of the FL pre-trained model. The accuracy of this model is referred to as Match Acc.

## B    RELATED WORK

Current PFL methods can primarily be categorized into several major types: **meta-learning-based methods** Fallah et al. (2020); Acar et al. (2021), **model-regularization-based methods** T Dinh et al. (2020); Li et al. (2021a), **fine-tuning-based methods** Jin et al. (2022); Chen et al. (2023); Li et al. (2023b), **personalized-weight-aggregation-based methods** Huang et al. (2021); Luo & Wu (2022), **feature-alignment-based methods** Xu et al. (2023); Zhou et al. (2024) and, **parameter-decoupling-based methods**. This paper delves into the issues inherent in the global model of FedAvg and primarily discusses parameter-decoupling methods that rely on the global model.

In addition to the aforementioned methods, a new category based on prompts has recently emerged.

**Prompt-based methods.**    Recently, prompt technology has garnered widespread attention in the fields of computer vision Jia et al. (2022); Liu et al. (2024) and natural language processing Lester et al. (2021); Liu et al. (2021). This technology involves using prompts as inputs to guide the behavior or output of models, typically for fine-tuning purposes. The domain of PFL has also seen the emergence of prompt-based approaches. Most of these are based on pre-trained models, aiming to train prompts to fine-tune the pre-trained models to fit client-local data, as seen in pFedPG Yang et al.

(2023), SGPT Deng et al. (2023), FedOTP Li et al. (2024), and FedAPT Su et al. (2024). pFedPT Li et al. (2023a) trains both the model and prompts, using prompts at the input level to learn personalized knowledge for fine-tuning the global model to adapt to the client's local distributions. Our FedPFT fundamentally differs from these methods in its objective. Rather than fine-tuning, we introduce prompts to guide feature transformations to align with the global classifier, thereby addressing the mismatch issue inherent in the global model during the training process.

## C  DETAILS OF COMBINING FEDPFT WITH CONTRASTIVE LEARNING

Following Section 3.6, this section provides details of the combination of FedPFT with contrastive learning, including the definition of $L_{\text{Con}}$ and the optimization process of Objective (6).

### C.1  DEFINITION OF $L_{\text{CON}}$

We adopt the Momentum Contrast (MoCo) framework He et al. (2020) for contrastive learning. The associated contrastive loss function is defined as:

$$L_{\text{Con}}(\phi, \tau, p_{\rho,i}, h_{\rho}; x) = -\log \frac{\exp\left(q \cdot k_{+}/\beta\right)}{\sum_{j=0}^{K} \exp\left(q \cdot k_j/\beta\right)}, \text{where } x \sim d_i. \tag{7}$$

In this formula, $h_{\rho}$ is the projection head used for contrastive learning. $q = h_{\rho} \circ \tau([\phi(x'), p_{\rho,i}])$ represents the query vector, and $k_{+} = \tilde{h}_{\rho} \circ \tau([\tilde{\phi}(x''), p_{\rho,i}])$ denotes the positive key vector. Here, $x'$ and $x''$ are augmented versions of the sample $x$, $\tilde{\phi}$ and $\tilde{h}_{\rho}$ refer to the momentum-updated encoder and projection head, respectively. $\beta$ is a temperature hyperparameter, and $K$ is the number of negative samples drawn from MoCo's queue, comprising the set $\{k_j\}_{j=0}^{K}$.

### C.2  OPTIMIZATION PROCESS OF FEDPFT+CON

As discussed in Section 3.5, FedPFT employs an alternating training strategy. In FedPFT+Con, we extend this approach by incorporating contrastive learning into the optimization process.

**Feature transformation phase.** In this phase, FedPFT+Con additionally utilizes $L_{\text{Con}}$ to update $\phi_i$, $\tau_i$ and $h_{\rho,i}$ to enhance the feature quality. The objective can be formulated as:

$$\min_{p_{\kappa,i}, \tau_i, \phi_i, h_{\rho,i}} \left\{ L_{\text{CE}}(\tau_i, p_{\kappa,i}; \phi_i, h_{\kappa}, d_i) + L_{\text{Con}}(\phi_i, \tau_i, h_{\rho,i}; p_{\rho,i}, d_i) \right\}. \tag{8}$$

**Model training phase.** FedPFT+Con additionally updates $p_{\rho,i}$ in this phase to align the features with the contrastive learning task, reducing interference from the classification task. Its training objective can be formulated as:

$$\min_{\phi_i, \tau_i, h_{\kappa,i}, p_{\rho,i}} \left\{ L_{\text{CE}}(\phi_i, \tau_i, h_{\kappa,i}; p_{\kappa,i}, d_i) + L_{\text{Con}}(p_{\rho,i}, \tau_i; \phi_i, h_{\rho,i}, d_i) \right\}. \tag{9}$$

Figure 7 illustrates the training process of the contrastive learning and classification tasks in FedPFT+Con. The blue modules represent components from FedPFT, while the orange modules represent additional components introduced in FedPFT+Con. Solid arrows indicate forward propagation and dashed arrows represent backpropagation.

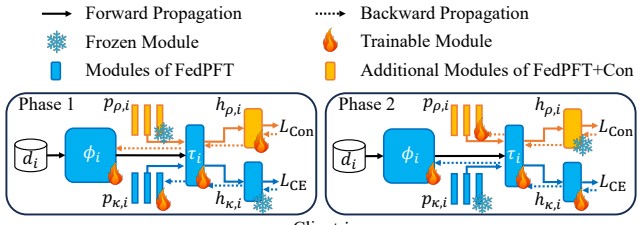

Figure 7: Training process of FedPFT+Con in each client $i$.

# D  EXPERIMENT SETUP

## D.1  INTRODUCTION TO NON-IID SCENARIOS

**Pathological non-IID.**    In this setting, each client is randomly assigned data from a subset of classes with equal data volume per class. For the CIFAR-10, CIFAR-100, and Tiny ImageNet datasets, we assign 2, 20, and 40 classes of data to each client, respectively.

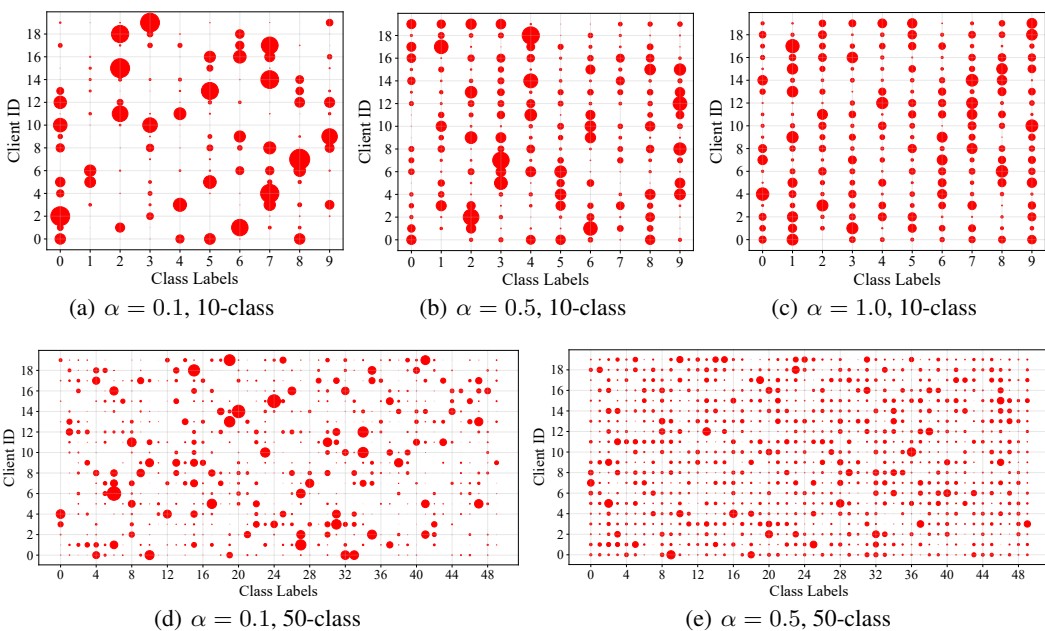

(a) $\alpha = 0.1$, 10-class          (b) $\alpha = 0.5$, 10-class          (c) $\alpha = 1.0$, 10-class

(d) $\alpha = 0.1$, 50-class                    (e) $\alpha = 0.5$, 50-class

Figure 8: Visualization of data partitioning in Dirichlet non-IID scenarios with different $\alpha$.

**Dirichlet non-IID.**    This is a commonly used setting in current FL research Wu et al. (2022; 2023); Shi et al. (2023). In this scenario, the data for each client is generated from a Dirichlet distribution denoted as $Dir(\alpha)$. As the value of $\alpha$ increases, the class imbalance within each client's dataset progressively decreases. This Dirichlet non-IID setting enables the evaluation of different methods across a broad spectrum of non-IID conditions, reflecting various degrees of data heterogeneity.

For a clearer, more intuitive understanding, we involve 20 clients with 10-class and 50-class datasets to visualize the data distribution among clients with varying $\alpha$ values. As depicted in Figure 8, the horizontal axis labels the data class indices, while the vertical axis lists the client IDs. Each red dot indicates the class data assigned to a client, with larger dots signifying a higher volume of data in that class.

## D.2  INTRODUCTION TO COMPARATIVE METHODS

FedAMP Huang et al. (2021) is a weighted-aggregation-based method where clients with similar data distributions are given higher aggregation weights during model aggregation. Because it mainly encourages the collaboration of clients with similar data distribution, it is a method that pays more attention to the local data distribution of clients. FedPer Arivazhagan et al. (2019), FedRep Collins et al. (2021), FedBN Li et al. (2021b), FedRoD Chen & Chao (2022), and FedCAC Wu et al. (2023) are parameter-decoupling-based methods, which personalize the global model by retaining certain parameters locally based on FedAvg. FedRoD additionally introduces a balanced global classifier to obtain assistance from other clients, alleviating the overfitting issue caused by personalized classifiers alone. pFedSD Jin et al. (2022) and pFedGate Chen et al. (2023) are fine-tuning-based methods that adapt the global model to local data through fine-tuning. pFedSD directly fine-tunes the global model by distilling local models, while pFedGate trains an additional gating network and applies it to the global model. pFedPT Li et al. (2023a), a prompt-based method, can also be viewed as a fine-tuning

approach, enhancing the global model's adaptation to local data distributions by adding prompts to images.

### D.3 HYPERPARAMETER SETTINGS IN DIFFERENT METHODS

For the unique hyperparameters of each baseline method, we utilize the optimal parameter combinations reported in their respective papers. For learning rates, we adjust within {1e-1, 1e-2, 1e-3}.

In FedPFT, to simplify the hyperparameter tuning process and enhance usability, we provide a default set of hyperparameters: for all scenarios, we set $n_\kappa = 10$ and $(R_f, R_a) = (4, 1)$. We use the SGD optimizer with a learning rate of 0.05 for the FTM and 0.1 for other components. In FedPFT+Con, for the Dirichlet non-IID scenario with $\alpha = 0.1$, we set $(R_f, R_a) = (3, 2)$, while in other scenarios, we use $(R_f, R_a) = (4, 1)$. The learning rate for the FTM remains 0.01, with other hyperparameters consistent with FedPFT. Unless otherwise specified, the above hyperparameter settings are used in our experiments, though fine-tuning these parameters for specific scenarios may yield better performance.

### D.4 COMPUTE RESOURCES

All the experiments are implemented using PyTorch and conducted on NVIDIA V100 GPUs. For the methods we compared, as well as FedPFT, a single training session requires 24-48 hours. For FedPFT+Con, the training process takes longer due to the use of the MoCo algorithm, which requires data augmentation that can only be executed on the CPU. Consequently, a single training session for FedPFT+Con requires 48-72 hours.

## E COMPARISON WITH STATE-OF-THE-ART METHODS

We present the comparative results of FedPFT and FedPFT+Con against established methods on CIFAR-10, CIFAR-100, and Tiny ImageNet datasets under Pathological non-IID scenarios, as well as CIFAR-10 under Dirichlet non-IID scenarios in Tables 7 and 8.

Table 7: Test accuracy (%) of different methods under Pathological non-IID setting on CIFAR-10, CIFAR-100, and Tiny ImageNet.

| Methods | CIFAR-10 | CIFAR-100 | Tiny ImageNet |
|---|---|---|---|
| FedAvg | $54.33 \pm 3.03$ | $34.27 \pm 0.44$ | $18.05 \pm 0.23$ |
| Local | $85.85 \pm 0.93$ | $38.40 \pm 0.69$ | $16.20 \pm 0.30$ |
| FedAMP | $88.88 \pm 0.83$ | $38.36 \pm 0.79$ | $16.13 \pm 0.55$ |
| FedPer | $87.51 \pm 0.95$ | $41.54 \pm 0.74$ | $20.25 \pm 0.65$ |
| FedRep | $87.10 \pm 0.91$ | $40.63 \pm 0.74$ | $19.24 \pm 0.33$ |
| FedBN | $87.02 \pm 1.41$ | $47.75 \pm 1.03$ | $24.91 \pm 0.48$ |
| FedRoD | $88.06 \pm 1.70$ | $52.55 \pm 0.92$ | $32.25 \pm 0.80$ |
| pFedSD | $89.97 \pm 1.45$ | $52.30 \pm 1.18$ | $30.27 \pm 0.78$ |
| pFedGate | $89.15 \pm 0.76$ | $43.73 \pm 0.14$ | $22.42 \pm 0.83$ |
| FedCAC | $89.77 \pm 1.14$ | $49.07 \pm 0.87$ | $30.83 \pm 0.42$ |
| pFedPT | $86.29 \pm 1.11$ | $39.92 \pm 0.33$ | $21.38 \pm 0.98$ |
| FedPFT | $89.67 \pm 1.96$ | $\mathbf{57.62 \pm 1.18}$ | $\mathbf{36.13 \pm 1.32}$ |
| FedPFT+Con | $\mathbf{90.55 \pm 1.35}$ | $\mathbf{58.14 \pm 0.71}$ | $\mathbf{37.59 \pm 0.39}$ |

**Results in Pathological non-IID scenario.** This is an extreme setting where each client has data from only a subset of classes. This scenario is particularly pronounced in the CIFAR-10 dataset, where each client essentially performs a simple binary classification task. Here, clients can achieve decent performance by solely focusing on their local tasks ('Local'), even without collaboration with other clients. As such, methods that prioritize local data distribution, such as FedAMP, pFedSD, and pFedGate, perform well. In contrast, on CIFAR-100 and Tiny ImageNet datasets, as clients have more local classes with fewer samples per class, local tasks become more challenging. Effective collaboration with other clients becomes crucial. Consequently, methods such as FedRoD, which

Table 8: Test accuracy (%) of different methods under Dirichlet non-IID setting on CIFAR-10.

| Methods | $\alpha = 0.1$ | $\alpha = 0.5$ | $\alpha = 1.0$ |
|---------|----------------|----------------|----------------|
| FedAvg | $60.39 \pm 1.46$ | $60.41 \pm 1.36$ | $60.91 \pm 0.72$ |
| Local | $81.91 \pm 3.09$ | $60.15 \pm 0.86$ | $52.24 \pm 0.41$ |
| FedAMP | $84.99 \pm 1.82$ | $68.26 \pm 0.79$ | $64.87 \pm 0.95$ |
| FedPer | $84.43 \pm 0.47$ | $68.80 \pm 0.49$ | $64.92 \pm 0.66$ |
| FedRep | $84.59 \pm 1.58$ | $67.69 \pm 0.86$ | $60.52 \pm 0.72$ |
| FedBN | $83.55 \pm 2.32$ | $66.79 \pm 1.08$ | $62.20 \pm 0.67$ |
| FedRoD | $86.23 \pm 2.12$ | $72.34 \pm 1.77$ | $68.45 \pm 1.94$ |
| pFedSD | $86.34 \pm 2.61$ | $71.97 \pm 2.07$ | $67.21 \pm 1.89$ |
| pFedGate | $87.25 \pm 1.91$ | $71.98 \pm 1.61$ | $67.85 \pm 0.87$ |
| FedCAC | $86.82 \pm 1.18$ | $69.83 \pm 0.46$ | $65.39 \pm 0.51$ |
| pFedPT | $82.38 \pm 2.91$ | $67.33 \pm 1.33$ | $64.37 \pm 1.22$ |
| FedPFT | $87.23 \pm 2.69$ | $\mathbf{74.10 \pm 1.95}$ | $\mathbf{69.23 \pm 0.76}$ |
| FedPFT+Con | $\mathbf{88.60 \pm 2.19}$ | $\mathbf{77.54 \pm 1.88}$ | $\mathbf{74.81 \pm 0.77}$ |

emphasize client collaboration, exhibit increasingly significant performance. FedAMP and pFedGate show considerable performance degradation. FedPer, FedRep, FedBN, and FedCAC, by personalizing certain parameters of FedAvg, enhance local performance by indirectly aligning local features with classifiers to some extent. However, as they do not address the mismatch issue, they compromise the performance of feature extractors to some extent, thereby limiting their performance to a moderate level across the three datasets. FedPFT aligns local features with the global feature space using classification prompts, enhancing both local feature-classifier alignment and inter-client collaboration effectiveness. It achieves competitive performance on CIFAR-10 and surpasses existing SOTA methods on CIFAR-100 and Tiny ImageNet. FedPFT+Con further incorporates contrastive learning tasks to enhance feature extractor performance, outperforming SOTA methods significantly across all datasets.

## F ADDRESSING MISMATCH BY INSERTING LINEAR LAYER

As discussed in Section 1, a straightforward approach to address the mismatch problem is to insert a personalized linear transformation layer between the global feature extractor and the global classifier (FedAvg+Linear). In this section, we validate this method through experiments, with the results shown in Table 9.

By combining the results in Table 2 and Table 8, we observe that FedAvg+Linear outperforms most SOTA methods on the CIFAR-10 dataset, demonstrating the effectiveness of addressing the mismatch problem during training. However, on the more challenging CIFAR-100 dataset, FedAvg+Linear underperforms several SOTA methods. This illustrates that a simple linear transformation is insufficient for complex datasets. Notably, on CIFAR-100 with $\alpha = 1.0$, FedAvg+Linear even underperforms FedAvg, highlighting that FedAvg+Linear tends to overfit the limited local training data due to the large number of personalized parameters introduced.

In comparison, FedPFT demonstrates superior performance across all scenarios. Leveraging the flexibility of the FTM, FedPFT+Con further enhances model performance, significantly outperforming FedAvg+Linear.

## G COMPARISON OF FEDPFT+CON WITH TWO-STAGE APPROACH

In FedPFT+Con, we propose an FTM to coordinate the joint training of contrastive learning and classification tasks. To illustrate the superiority of this design, we introduce a baseline called 'Two-stage,' similar to Wang et al. (2023), where contrastive learning training is conducted first, followed by classification task training after convergence. For fairness, in the two-stage method, we first

Table 9: Test accuracy (%) of FedAvg+Linear under Dirichlet non-IID on CIFAR-10 and CIFAR-100.

| | CIFAR-10 | | | CIFAR-100 | | |
|---|---|---|---|---|---|---|
| Methods | $\alpha = 0.1$ | $\alpha = 0.5$ | $\alpha = 1.0$ | $\alpha = 0.1$ | $\alpha = 0.5$ | $\alpha = 1.0$ |
| FedAvg | 60.39±1.46 | 60.41±1.36 | 60.91±0.72 | 34.91±0.86 | 32.78±0.23 | 33.94±0.39 |
| Local | 81.91±3.09 | 60.15±0.86 | 52.24±0.41 | 47.61±0.96 | 22.65±0.51 | 18.76±0.63 |
| FedAvg+Linear | 85.96±2.23 | 71.17±1.28 | 67.63±0.83 | 58.07±0.41 | 37.09±0.85 | 31.23±0.24 |
| FedPFT | **87.23±2.69** | **74.10±1.95** | **69.23±0.76** | **60.98±0.39** | **44.87±0.76** | **41.83±0.37** |
| FedPFT+Con | **88.60±2.19** | **77.54±1.88** | **74.81±0.77** | **62.03±1.41** | **47.98±0.78** | **44.29±0.74** |

perform 1000 rounds of contrastive learning training, followed by 1000 rounds of classification task training. The experimental results are depicted in Figure 9.

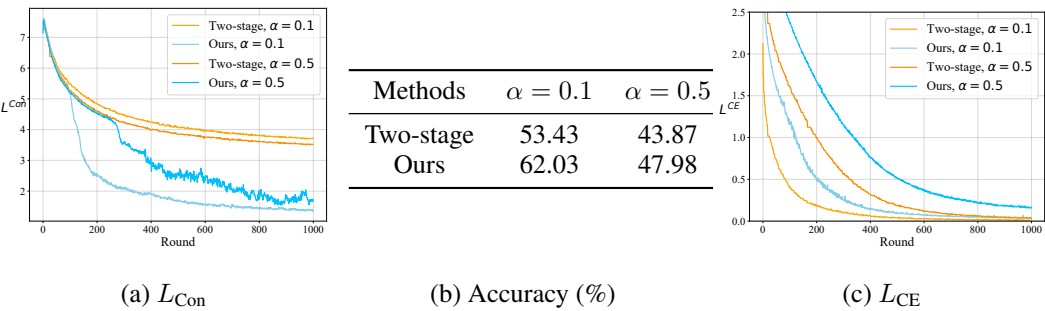

| Methods | $\alpha = 0.1$ | $\alpha = 0.5$ |
|---|---|---|
| Two-stage | 53.43 | 43.87 |
| Ours | 62.03 | 47.98 |

(a) $L_{\text{Con}}$     (b) Accuracy (%)     (c) $L_{\text{CE}}$

Figure 9: Comparison with two-stage approach on training $L_{\text{Con}}$, $L_{\text{CE}}$, and testing accuracy.

Firstly, from the perspective of the contrastive learning loss ($L_{\text{Con}}$), FedPFT+Con registers lower loss values compared to the Two-stage approach, suggesting that simultaneous training with the classification task enhances the efficacy of contrastive learning. Secondly, considering both Figure 9(b) and Figure 9(c), our method exhibits significantly higher accuracy compared to the Two-stage approach. However, $L_{\text{CE}}$ converges to a higher training loss value, suggesting that in our design, contrastive learning tasks can alleviate overfitting issues in the classification task during training. These experiments demonstrate that our proposed approach can effectively coordinate both tasks, allowing them to assist each other. Importantly, these experiments also indicate that the significant performance improvement brought by contrastive learning in our method is largely attributed to the design of our FTM and training approach.

## H  ATTENTION WEIGHT VISUALIZATION

In the FTM of FedPFT and FedPFT+Con, self-attention mechanisms are employed to facilitate the integration of prompts with sample features. This section visualizes the attention weights to reveal how prompts influence the transformation process. We analyze 20 test samples from a single client on the CIFAR-10 dataset, with results depicted in Figure 10. Each row in the figure corresponds to the attention weights for the output feature $f'$ of a single sample. Columns represent the input dimensions of the FTM: the first column corresponds to the original input feature $f$, while subsequent columns relate to different prompts from the sets $p_{\kappa,i}$ or $p_{\rho,i}$.

It can be observed that when $\alpha = 0.1$, indicating severe local class imbalances, each client has data from only a few classes. In this case, the feature transformation task is relatively simple, and the influence of different prompts on a sample is similar. As $\alpha$ increases, indicating more complex local tasks, the influence of prompts becomes more intricate. Particularly at $\alpha = 1.0$, it can be seen that each sample is affected differently by different prompts. This also indicates that our approach performs sample-level feature transformation.

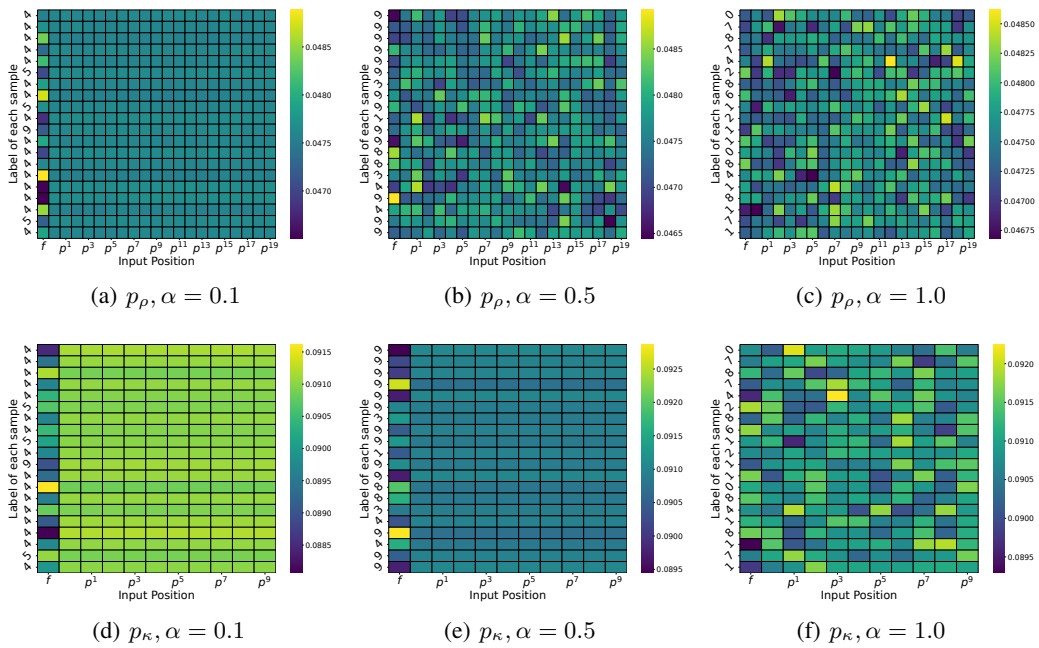

Figure 10: Visualize attention weights for different prompts in a client in the CIFAR-10 dataset under the Dirichlet non-IID scenario.

# I   EXPERIMENTS IN FEATURE SHIFT NON-IID SCENARIOS

In the experiments presented in our main text, we primarily conduct validation in the label shift non-IID scenarios. In this section, we further validate FedPFT in feature shift non-IID scenarios.

We conduct experiments on two feature shift datasets, PACS Li et al. (2017) and DomainNet Peng et al. (2019). PACS and DomainNet have four and six domains, respectively, with each domain assigned to one client. The number of clients corresponds to the number of domains. For each client, we allocate 1000 training samples and 500 testing samples. The experimental results are shown in Table 10.

Table 10: Comparison of different methods in the feature shift non-IID scenarios.

| Methods | PACS | | DomainNet | |
|---|---|---|---|---|
| | Origin Acc. | Match Acc. | Origin Acc. | Match Acc. |
| FedAvg | 71.48% | 75.49% | 63.53% | 67.17% |
| FedPer | 74.86% | 75.31% | 65.70% | 65.67% |
| FedBN | 73.91% | 74.71% | 67.57% | 68.57% |
| FedCAC | 74.94% | 75.94% | 67.80% | 68.53% |
| FedPFT | 77.67% | 77.64% | 70.37% | 70.57% |

**Existence of the mismatch phenomenon.** The results show that in both datasets, there are still noticeable gaps between Origin Acc. and Match Acc. across all methods, especially in FedAvg. This indicates that the mismatch problem persists and is a major reason for the suboptimal performance of FedAvg.

**Superiority of our method.** From the perspective of mismatch degree, the small gap between Origin Acc. and Match Acc. in FedPFT demonstrates its effectiveness in addressing the mismatch problem in feature shift non-IID scenarios. In terms of Origin Acc., FedPFT significantly outperforms SOTA methods (e.g., by up to 2.73% on PACS), further highlighting the superiority of FedPFT in address the feature shift non-IID problem.

## J    EXPANDING TO MORE CLIENTS AND MORE COMPLEX MODELS

To demonstrate the scalability of FedPFT, we further conduct experiments on CIFAR-10 in Dirichlet non-IID scenarios with more clients and more complex models.

**Scaling to more clients.** We perform experiments with 100 clients, and the test accuracies of various methods are presented in Table 11.

Table 11: Accuracy (%) of different methods with 100 clients.

| Scenarios | FedPer | FedBN | FedRoD | FedCAC | FedPFT |
|---|---|---|---|---|---|
| $\alpha = 0.1$ | 84.68 | 85.51 | 87.58 | 87.40 | **88.43** |
| $\alpha = 0.5$ | 71.40 | 70.85 | 75.23 | 72.82 | **76.49** |
| $\alpha = 1.0$ | 66.92 | 67.18 | 70.99 | 69.06 | **72.53** |

**Scaling to more complex models.** We conduct experiments using ResNet-18, with the test accuracies of different methods shown in Table 12.

Table 12: Accuracy (%) of different methods with ResNet-18.

| Scenarios | FedPer | FedBN | FedRoD | FedCAC | FedPFT |
|---|---|---|---|---|---|
| $\alpha = 0.1$ | 82.80 | 78.10 | 83.50 | 82.60 | **83.52** |
| $\alpha = 0.5$ | 65.35 | 55.38 | 67.55 | 63.27 | **68.93** |
| $\alpha = 1.0$ | 61.83 | 56.88 | 62.05 | 60.53 | **64.95** |

In both experiments, FedPFT significantly outperforms state-of-the-art methods, highlighting the scalability of our approach.

## K    PARTIAL CLIENT PARTICIPATION

In FL, challenges such as offline clients and unstable communication may result in only a subset of clients participating in training each round, posing a challenge to the robustness of FL algorithms. In this section, we investigate whether FedPFT is robust to this issue. We conduct experiments on CIFAR-10, CIFAR-100, and Tiny ImageNet, considering scenarios where only a random 50%, 70%, and 90% of clients participate in training each round. The experimental results are presented in Table 13.

Table 13: Accuracy (%) of FedPFT when different proportions of clients participate in each round of training. The content in '()' represents the performance change compared to 100% client participation.

| Datasets | 100% | 90% | 70% | 50% |
|---|---|---|---|---|
| CIFAR-10 | 74.10±1.95 | 73.88±1.84 (-0.22) | 74.21±1.45 (+0.11) | 74.33±1.38 (+0.23) |
| CIFAR-100 | 44.87±0.76 | 45.74±0.32 (+0.87) | 45.46±1.14 (+0.59) | 45.87±0.81 (+1.00) |
| Tiny | 28.61±0.40 | 28.53±0.62 (-0.08) | 29.24±0.16 (+0.63) | 29.90±0.10 (+1.29) |

It can be observed that compared to scenarios where all clients participate in training, FedPFT's accuracy is not significantly reduced when only a subset of clients participate. Furthermore, in CIFAR-100 and Tiny ImageNet, the performance of FedPFT may even be improved. This is because reducing the number of participating clients each round may mitigate the impact of non-IID data distribution on the global model. These experiments demonstrate the robustness of FedPFT to scenarios where only a subset of clients participate.

## L    EFFECT OF HYPERPARAMETERS

In the previous experiments, we utilize the default hyperparameter combination. In this section, we verify how variations in these hyperparameters influence the performance of FedPFT.

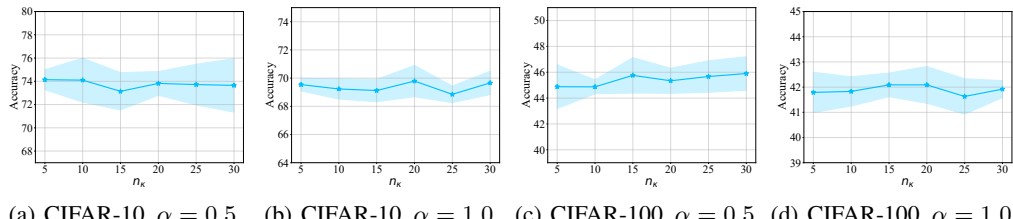

(a) CIFAR-10, $\alpha = 0.5$    (b) CIFAR-10, $\alpha = 1.0$    (c) CIFAR-100, $\alpha = 0.5$    (d) CIFAR-100, $\alpha = 1.0$

Figure 11: The effect of hyperparameter $n_\kappa$ on CIFAR-10 and CIFAR-100 in the Dirichlet non-IID scenario.

### L.1 EFFECT OF $n_\kappa$

$n_\kappa$ represent the number of prompts in $p_{\kappa,i}$ for each client. We examine the impact of this hyperparameter on the performance of FedPFT on CIFAR-10 and CIFAR-100 datasets. The experimental results are depicted in Figure 11.

FedPFT shows considerable robustness to variations in $n_\kappa$. On the CIFAR-10 dataset, changes in $n_\kappa$ have minimal impact on performance, suggesting that the model can effectively handle simpler data distributions even with fewer prompts. In contrast, on the more complex CIFAR-100 dataset, performance is initially limited by a small number of prompts, which may not sufficiently cover the diverse feature space required for effective feature transformation. As the number of prompts increases, the model's ability to transform and adapt features improves, leading to enhanced performance.

### L.2 EFFECT OF $R_f$ AND $R_a$

$R_f$ and $R_a$ are used to control the number of training epochs for the two training stages. Since we set $R_f + R_a = R$, in this experiment, we only adjust $R_f$ to examine the impact of these two hyperparameters on model performance. The experimental results are illustrated in Figure 12.

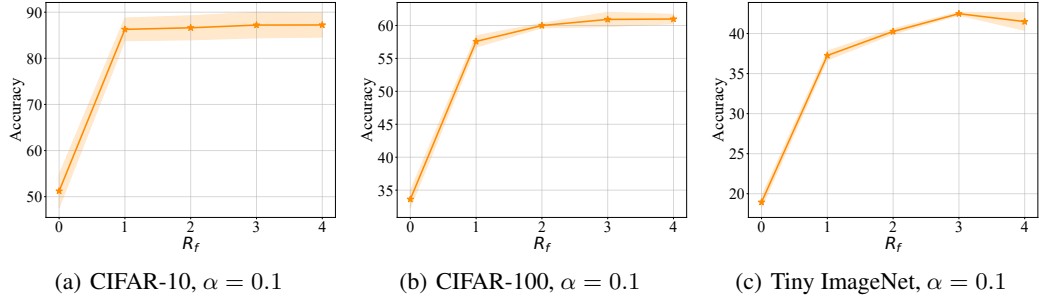

(a) CIFAR-10, $\alpha = 0.1$        (b) CIFAR-100, $\alpha = 0.1$        (c) Tiny ImageNet, $\alpha = 0.1$

Figure 12: The effect of hyperparameter $R_f$ on CIFAR-10, CIFAR-100, and Tiny ImageNet in the Dirichlet non-IID scenario with $\alpha = 0.1$.

When $R_f = 0$, it indicates that local features are not aligned with the global classifier before training the model parameters. Under this condition, the model performance is observed to be very poor. As $R_f$ gradually increases, the model performance initially improves but then declines in some scenarios, suggesting that $R_f$ balances the trade-off between the two training stages. When $R_f$ is too small, local features are not sufficiently transformed to match the classifier, resulting in the model being affected by the mismatch during the model training phase, which reduces the synergy between the feature extractor and classifier. On the other hand, when $R_f$ is too large, the model parameters are insufficiently trained, limiting the learning of local knowledge from clients.

In general, $R_f$ and $R_a$ are two hyperparameters that need careful adjustment, as they have a significant impact on the performance of FedPFT. Typically, in scenarios where clients' local tasks are simple, it

may be appropriate to decrease the value of $R_f$. In other cases, we recommend using a larger $R_f$ value to fully align the local features with the global classifier.

## M    COMPUTATION EFFICIENCY

We empirically evaluate the computational efficiency of our method on CIFAR-100 using ResNet-10, with results displayed in Table 14. We run each method for 100 rounds and calculate their average runtime per round. Each method exclusively utilizes a single machine during runtime. All experiments are conducted on four NVIDIA RTX 2080 GPUs.

Table 14: The average computation time per round for different methods on CIFAR-100.

| Methods | FedAvg | FedPer | FedBN | FedRoD | FedCAC | FedPFT |
|---|---|---|---|---|---|---|
| Time per round (s) | 50.10 | 51.81 | 52.65 | 54.76 | 55.35 | 51.61 |

FedPFT introduces the FTM, which adds some computational overhead. However, since it does not require updating the feature extractor during the feature learning phase, this helps reduce training costs to some extent. Overall, from the results in Table 14, FedPFT takes slightly longer to compute than FedAvg but is more time-efficient compared to SOTA methods.

## N    COMMUNICATION COST

In this section, we calculate the communication overhead of one client in FedAvg and FedPFT in each communication round.

Table 15: The communication cost of each client in FedAvg and FedPFT in one communication round. The percentages in parentheses represent the increase compared to FedAvg.

| Models | $\phi_i$ | $\tau_i$ | $h_{\kappa,i}$ | FedAvg | FedPFT |
|---|---|---|---|---|---|
| ResNet-8 | 1.24M | 0.26M | 25.70K | 1.27M | 1.53M (20.47%) |
| ResNet-10 | 4.91M | 1.05M | 51.30K | 4.96M | 6.01M (21.17%) |

In FedAvg, each communication round uploads the feature extractor $\phi_i$ and classifier $h_{\kappa,i}$. FedPFT adds FTM $\tau_i$, increasing communication overhead by 20.47% for ResNet-8 and 21.17% for ResNet-10.

While FedPFT brings additional communication costs, it is important to weigh it against the performance enhancements and flexibility offered by $\tau_i$, as discussed in earlier sections of this paper. The improved model accuracy and robustness to non-IID data might justify the additional costs in scenarios where model performance is critical.

Moving forward, considering the increase in communication cost is primarily due to the additional components $\tau_i$, we aim to develop a more efficient and lightweight FTM to reduce communication demands without compromising model effectiveness in our future work.

## O    LIMITATIONS AND FUTURE WORK

In this paper, we primarily investigate PFL methods that derive personalized models based on a global model. We analyze the essential reasons these methods enhance performance from the perspective of mismatches between local features and classifiers. Although such methods occupy the mainstream in the current PFL field, it is necessary to admit that there are some PFL methods that are not based on global models, such as personalized-weight-aggregation-based methods, which are not explored in this study. Additionally, while this paper observes that personalizing a subset of parameters degrades the quality of the feature extractor, the underlying reasons for this phenomenon require further investigation.

# P  THEORETICAL ANALYSIS

Since the main problem in Eq. (1) is non-convex, we focus on the factors affecting convergence in the non-convex setting.

Table 16: The glossary of notations used in the theoretical analysis.

| Implication | Notation |
|---|---|
| Global / Local loss | $L$ / $L_i$ |
| Global / Local problem | $F$ / $F_i$ |
| Local Dataset on $i^{\text{th}}$ client | $\tilde{d}_i \in d_i$ |
| Feature extractor | $\phi$ |
| Feature transformation module | $\tau$ |
| Classification / Contrastive learning prompts | $p_\kappa$ / $p_\rho$ |
| Feature extractor & Feature transformation module & Classifier | $w$ |
| Classification / Contrastive learning task head | $h_\kappa$ / $h_\rho$ |
| Global / Local problem's gradient | $\nabla F(w)$ / $\nabla F_i(w)$ |
| Local gradient approximation | $g_{i,r}^t$ |
| Client number | $N$ |
| Local update epoch | $R$ |
| The number of clients sampled at each global epoch | $S$ |
| The set of clients sampled at global epoch $t$ | $\mathcal{S}_t$ |
| The actual learning rate of global problem | $\tilde{\eta}$ |
| The learning rate of local problem | $\eta$ |
| Approximated local gradient error's upper-bound | $\delta$ |
| Local-global gradient error's upper-bound | $\sigma_F$ |
| Index of client, local epoch and global epoch | $i \in [N], r \in [R], t \in [T]$ |

## P.1  PROBLEM SETUP

Non-convex case analyses are provided, because our model is multi-layer transformer. Analyses are as follows.

We transform the problem into an unconditional bi-level optimization problem:

$$\min_w \mathbf{E} F(w) = \mathbf{E}_i \{ F_i(w) := \min_{p_{\kappa,i}} \mathbf{E}_{d_i} L_{\text{CE}}(w, p_{\kappa,i}; d_i) \}$$

where $\mathbf{E}$ represents the expectation of all random variables, $\mathbf{E}_i$ means the expectation of client sampling, $\mathbf{E}_{d_i}$ is the local data sampling expectation, and we use $w = \{\phi, \tau, h_\kappa\}$ for simplification, based on the equivalence of block coordinate descent and gradient descent.

With contrastive learning the problem could be transformed into a similar problem with constrain. By Lagrange duality, the main problem is transformed as follows:

$$\min_{\phi, \tau, h_\kappa} \min_{\{p_{\kappa,i}\}_{i \in [N]}} \mathbf{E}_i \mathbf{E}_{d_i} L_{\text{CE}}(\phi, \tau, h_\kappa, p_{\kappa,i}; d_i)$$

$$\text{s.t. } \mathbf{E}_i \mathbf{E}_{d_i} L_{\text{Con}}(\phi, \tau; d_i) \leq H_{\text{Con}}$$

## P.2  PROPOSITIONS

**Proposition P.1** (*L*-smooth). *If $f$ is $L$-smooth, $\forall x, y$ we have:*

$$\langle \nabla f(x) - \nabla f(y), x - y \rangle \leq L ||x - y||^2$$

$$||\nabla f(x) - \nabla f(y)|| \leq L ||x - y||$$

$$||\nabla f(x) - \nabla f(y)||^2 \leq 2L[f(x) - f(y)]$$

$$f(y) - f(x) - \langle \nabla f(x), y - x \rangle \leq \frac{L}{2} ||y - x||^2$$

**Proposition P.2** (Jensen's inequality). *If $f$ is convex, we have the following inequality:*

$$\mathbf{E}_X f(X) \geq f(\mathbf{E}_X X).$$

*A variant of the general one shown above, given a group $\{x_i\}_{i \in [N]}$:*

$$|| \sum_{i \in [N]} x_i||^2 \leq N \sum_{i \in [N]} ||x_i||^2.$$

**Proposition P.3** (Triangle inequality). *The triangle inequality, where $|| \cdot ||$ is the norm, and $A$, $B$ is the elements in the corresponding norm space:*

$$||A + B|| \leq ||A|| + ||B||$$

**Proposition P.4** (Matrix norm compatibility). *The matrix norm compatibility, $A \in \mathbf{R}^{a \times b}, B \in \mathbf{R}^{b \times c}, v \in \mathbf{R}^b$:*

$$||AB||_m \leq ||A||_m ||B||_m$$
$$||Av||_m \leq ||A||_m ||v||$$

**Proposition P.5** (Peter Paul inequality). *$\forall x, y$ and $\forall \epsilon > 0$, we have the following inequality:*

$$2\langle x, y \rangle \leq \frac{1}{\epsilon} ||x||^2 + \epsilon ||y||^2$$

## P.3 Assumptions

**Assumption P.1** (L-smooth local objectives). *$\forall i$, $F_i$ is $L_F$-Smooth, the main proposition is shown in Prop. P.1. Notice that the $F_i$ is assumed to be L-smooth and non-convex, which matches the problem and neural network architecture setting in the main paper.*

**Assumption P.2** (Bounded local variance). *The local problem's gradient is assumed not to be too far from the global problem's gradient.*

$$\forall w, \mathbf{E}_i ||\nabla F_i(w) - \nabla F(w)|| \leq \sigma_F$$

**Assumption P.3** (Bounded approximated gradient). *The first-order approximation of the local problem's gradient $g_{i,r}^t$ should not be too far from the ground truth $\nabla F_i(w_{i,r}^t)$. In this assumption, the approximated error of the block coordinate descent in Algorithm 1 is bounded.*

$$\forall \{(i, r, t)\}, ||g_{i,r}^t - \nabla F_i(w_{i,r}^t)|| \leq \delta$$

## P.4 Lemmas

**Lemma P.1** (Bounded local approximation error). *If $\tilde{\eta} := \eta R \leq \frac{1}{2L_F}$, we have the following bound of client drift error:*

$$\frac{1}{NR} \sum_{i,r}^{N,R} \mathbf{E} ||g_{i,r}^{(t)} - \nabla F_i(w^{(t)})||^2 \leq 2\delta^2 + 2^{R+3} L_F [3\tilde{\eta}^2 \sum_i^N \mathbf{E} ||\nabla F_i(w^{(t)})||^2 + \frac{2\tilde{\eta}^2 \delta^2}{R}]$$

*Proof.* The client drift error on given $i^{\text{th}}$ client and its upper bound are as follows:

$$\begin{aligned}
&\mathbf{E} ||g_{i,r}^{(t)} - \nabla F_i(w^{(t)})||^2 \\
\leq &2\mathbf{E} ||g_{i,r}^{(t)} - \nabla F_i(w_{i,r}^{(t)})||^2 + 2\mathbf{E} ||\nabla F_i(w^{(t)}) - \nabla F_i(w_{i,r}^{(t)})||^2 \\
\leq &2\delta^2 + 2L_F \mathbf{E} ||w_{i,r}^{(t)} - w^{(t)}||^2
\end{aligned} \quad (10)$$

where the first inequality is by Proposition P.3 and the second one is by Assumption P.1.

For the last term in the upper bound, we have the iterative formulation as follows:

$$
\begin{aligned}
&\mathbf{E}||w_{i,r}^{(t)} - w^{(t)}||^2 \\
=&\mathbf{E}||w_{i,r-1}^{(t)} - w^{(t)} - g_{i,r-1}^{(t)}||^2 \\
\leq&2\mathbf{E}||w_{i,r-1}^{(t)} - w^{(t)} - \eta\nabla F_i(w^{(t)})||^2 + 2\eta^2\mathbf{E}||g_{i,r-1}^{(t)} - \nabla F_i(w^{(t)})||^2 \\
\leq&2(1 + \frac{1}{2R})\mathbf{E}||w_{i,r-1}^{(t)} - w^{(t)}||^2 + 2(1 + 2R)\eta^2\mathbf{E}||\nabla F_i(w^{(t)})||^2 \\
&+ 4\eta^2[\delta^2 + L_F^2\mathbf{E}||w_{i,r}^{(t)} - w^{(t)}||^2] \\
=&2(1 + \frac{1}{2R} + 2\eta^2 L_F^2)\mathbf{E}||w_{i,r-1}^{(t)} - w^{(t)}||^2 + 4\eta^2\delta^2 \\
&+ 2(1 + 2R)\eta^2\mathbf{E}||\nabla F_i(w^{(t)})||^2
\end{aligned}
$$

where the two inequalities are by Proposition P.3, Proposition P.5 and Eq. (10).

Take $\tilde{\eta} := \eta R \leq \frac{1}{2L_F}$, we recursively unroll the inequality as follows:

$$
\begin{aligned}
&\mathbf{E}||w_{i,r}^{(t)} - w^{(t)}||^2 \\
\leq&2(1 + \frac{1}{R})\mathbf{E}||w_{i,r-1}^{(t)} - w^{(t)}||^2 + 4\eta^2\delta^2 + 2(1 + 2R)\eta^2\mathbf{E}||\nabla F_i(w^{(t)})||^2 \\
\leq&[3\tilde{\eta}^2\mathbf{E}||\nabla F_i(w^{(t)})||^2 + \frac{2\tilde{\eta}^2\delta^2}{R}]2^{R+2}
\end{aligned}
$$

where the inequality is unrolled and we use $\frac{1}{R} \leq 1$. Thus, we have:

$$
\mathbf{E}||g_{i,r}^{(t)} - \nabla F_i(w^{(t)})||^2 \leq 2\delta^2 + 2^{R+4}\tilde{\eta}^2 L_F[3\sigma_F^2 + 3\mathbf{E}||\nabla F(w^{(t)})||^2 + \frac{\delta^2}{R}]
$$

$\square$

P.5   THEOREM AND DISCUSSION

**Theorem P.2** (Non-convex and smooth convergence of FedPFT). *Let Assumption P.1, Assumption P.2 and Assumption P.3 hold, if $\tilde{\eta} := \eta R \leq \min\{\frac{1}{2L_F}, \hat{\eta}\}$ is taken, where $\hat{\eta} := \frac{N/S-1}{24(N-1)2^R}\sigma_F^2 - 1$, we have the following bound:*

$$
\mathcal{O}(\mathbf{E}||\nabla F(w^{(\bar{t})})||^2) := \mathcal{O}(\frac{\Delta_F}{\hat{\eta}T} + \frac{2^{R/3}L_F^{1/3}(R\sigma_F^2 + \delta^2)^{1/3}\Delta_F^{2/3}}{T^{2/3}R^{1/3}} + (\frac{\sigma_F\sqrt{L_F(N/S - 1)\Delta_F}}{\sqrt{TN}}) + \delta^2)
$$

*Proof.*

$$\mathbf{E}F(w^{(t+1)}) - \mathbf{E}F(w^{(t)})$$

$$\leq \mathbf{E}\langle \nabla F(w^{(t)}), w^{(t+1)} - w^{(t)}\rangle + \frac{L_F}{2}\mathbf{E}||w^{(t+1)} - w^{(t)}||^2$$

$$= -\tilde{\eta}\mathbf{E}\langle \nabla F(w^{(t)}), g^{(t)}\rangle + \frac{\tilde{\eta}^2 L_F}{2}\mathbf{E}||g^{(t)}||^2$$

$$= -\tilde{\eta}\mathbf{E}||\nabla F(w^{(t)})||^2 - \tilde{\eta}\mathbf{E}\langle \nabla F(w^{(t)}), g^{(t)} - \nabla F(w^{(t)})\rangle + \frac{\tilde{\eta}^2 L_F}{2}\mathbf{E}||g^{(t)}||^2$$

$$\leq -\frac{\tilde{\eta}}{2}\mathbf{E}||\nabla F(w^{(t)})||^2 + \frac{\tilde{\eta}}{2}\mathbf{E}||\frac{1}{NR}\sum_{i,r}^{N,R}g_{i,r}^{(t)} - \nabla F_i(w^{(t)})||^2 + \frac{\tilde{\eta}^2 L_F}{2}\mathbf{E}||g^{(t)}||^2$$

$$\leq -\frac{\tilde{\eta}}{2}\mathbf{E}||\nabla F(w^{(t)})||^2 + \frac{\tilde{\eta}}{2}\mathbf{E}||\frac{1}{NR}\sum_{i,r}^{N,R}g_{i,r}^{(t)} - \nabla F_i(w^{(t)})||^2$$

$$+ \frac{3\tilde{\eta}^2 L_F}{2}\mathbf{E}[||g^{(t)} - \nabla F_i(w^{(t)})||^2 + ||\frac{1}{S}\sum_{i\in\mathcal{S}^{(t)}}\nabla F_i(w^{(t)}) - \nabla F(w^{(t)})||^2 + ||\nabla F(w^{(t)})||^2]$$

$$= -\frac{\tilde{\eta}(1 - 3\tilde{\eta}L_F)}{2}\mathbf{E}||\nabla F(w^{(t)})||^2 + \frac{\tilde{\eta}(1 + 3\tilde{\eta}L_F)}{2}\frac{1}{NR}\sum_{i,r}^{N,R}\mathbf{E}||g_{i,r}^{(t)} - \nabla F_i(w^{(t)})||^2$$

$$+ \frac{3\tilde{\eta}^2 L_F}{2}||\frac{1}{S}\sum_{i\in\mathcal{S}^{(t)}}\nabla F_i(w^{(t)}) - \nabla F(w^{(t)})||^2$$

$$\leq -\frac{\tilde{\eta}(1 - 3\tilde{\eta}L_F)}{2}\mathbf{E}||\nabla F(w^{(t)})||^2 + 3\tilde{\eta}^2 L_F\frac{N/S - 1}{N - 1}[\sigma_F^2 + ||\nabla F(w^{(t)})||^2]$$

$$+ \tilde{\eta}(1 + 3\tilde{\eta}L_F)[\delta^2 + 2^{R+3}\tilde{\eta}^2 L_F[3\sigma_F^2 + 3\mathbf{E}||\nabla F(w^{(t)})||^2 + \frac{\delta^2}{R}]]$$

where the four inequalities are respectively by $L_F$-smooth of $F := \mathbf{E}_i F_i$, Proposition P.5, Lemma P.1 and the similar classic Lemma 4 in (Shi et al., 2023).

Let $c_1 := 3\delta^2$, $c_2 := 3L_F\sigma_F^2\frac{N/S-1}{N-1}$, $c_3 := 2^{R+3}L_F[3\sigma_F^2 + \frac{\delta^2}{R}]$,

$$\mathbf{E}F(w^{(t+1)}) - \mathbf{E}F(w^{(t)}) \leq -\frac{\tilde{\eta}}{2}\{1 - [\frac{3}{2} - 3\frac{N/S - 1}{N - 1}\sigma_F^2 + 72 \times 2^R\tilde{\eta}]\}\mathbf{E}||\nabla F(w^{(t)})||^2$$

$$+ c_3\tilde{\eta}^3 + c_2\tilde{\eta}^2 + c_1\tilde{\eta}$$

$$\leq -\frac{\tilde{\eta}}{2}\mathbf{E}||\nabla F(w^{(t)})||^2 + c_3\tilde{\eta}^3 + c_2\tilde{\eta}^2 + c_1\tilde{\eta}$$

where let $\tilde{\eta} \leq \min\{\frac{1}{2L_F}, \hat{\eta}$, where $\hat{\eta} := \frac{2}{3\times 2^{R+4}}\frac{N/S-1}{N-1}\sigma_F^2 - 1\}$. Re-arranging the inequality above and accumulating, we have:

$$\frac{1}{2}\mathbf{E}||\nabla F(w^{(t)})||^2 \leq \mathbf{E}F(w^{(t+1)}) - \mathbf{E}F(w^{(t)}) + c_3\tilde{\eta}^2 + c_2\tilde{\eta} + c_1$$

$$\frac{1}{2T}\sum_{t=0}^{t=T-1}\mathbf{E}||\nabla F(w^{(t)})||^2 \leq \mathbf{E}F(w^{(T)}) - \mathbf{E}F(w^{(0)}) + c_3\tilde{\eta}^2 + c_2\tilde{\eta} + c_1$$

Let $\Delta_F = F(w^0) - F(w^*)$, where $w^*$ is the minimum of the main problem $\arg\min_w \mathbf{E}F(w)$. To measure the exact term of the bounds, we consider the following cases:

- $\frac{\Delta_F}{c_3 T} \leq \tilde{\eta}^3$ or $\frac{\Delta_F}{c_2 T} \leq \tilde{\eta}^2$, let $\tilde{\eta} = \min\{(\frac{\Delta_F}{c_3 T})^{1/3}, (\frac{\Delta_F}{c_2 T})^{1/2}\}$, we have:

$$\frac{1}{2}\mathbf{E}||\nabla F(w^{(t)})||^2 \leq \frac{c_3^{1/3}\Delta_F^{2/3}}{T^{2/3}} + (\frac{c_2\Delta_F}{T})^{1/2} + c_1$$

- $\frac{\Delta_F}{c_3 T} \geq \tilde{\eta}^3$ and $\frac{\Delta_F}{c_2 T} \geq \tilde{\eta}^2$, let $\tilde{\eta} = \hat{\eta}$, we have:

$$\frac{1}{2}\mathbf{E}||\nabla F(w^{(t)})||^2 \leq \frac{\Delta_F}{\hat{\eta} T} + \frac{c_3^{1/3}\Delta_F^{2/3}}{T^{2/3}} + (\frac{c_2\Delta_F}{T})^{1/2} + c_1$$

Uniformly sample a $\bar{t} \in [T] - 1$, we have the upper bound as follows:

$$\frac{1}{T}\sum_{t=0}^{T-1}\mathbf{E}||\nabla F(w^{(t)}||^2) = \mathcal{O}(\mathbf{E}||F(w^{(\bar{t})})||^2)$$

$$:= \mathcal{O}(\frac{\Delta_F}{\hat{\eta} T} + \frac{2^{R/3}L_F^{1/3}(R\sigma_F^2 + \delta^2)^{1/3}\Delta_F^{2/3}}{T^{2/3}R^{1/3}} + (\frac{\sigma_F\sqrt{L_F(N/S - 1)\Delta_F}}{\sqrt{TN}}) + \delta^2)$$

$\square$

**Remark P.2.1.** *According to Theorem P.2, our proposed FedPFT converges at a sub-linear level. The linear term $\mathcal{O}(\frac{\Delta_F}{\hat{\eta}T})$ is affected by $\hat{\eta}$ and the initialization gap $\Delta_F$. The sub-linear term $\mathcal{O}(1/T^{2/3})$ is affected by $R$, especially when $R$ is large due to the exponential factor $2^R$. As the local approximation error of the gradient $\delta$ grows, both the convergence radius $\mathcal{O}(\delta)$ and the sub-linear term $\mathcal{O}(1/T^{2/3})$ are affected by the local optimizer selection significantly. Another sub-linear term $\mathcal{O}(\sqrt{T})$ is eliminated if $N/S - 1 = 0$ when all the clients are sampled. Otherwise, the sub-linear rate is mainly affected by $\sigma_F$.*

*FedPFT aligns the training objectives across clients by introducing $p_{\kappa,i}$ . Our design can effectively reduce differences in local gradients among clients during training, thereby reducing $\sigma_F$ and subsequently lowering the upper bound. During training, $p_{\kappa,i}$ incorporate information from the local datasets. By using them as part of the input, FedPFT effectively reduces the randomness in gradient computation, thereby lowering $\delta$ and consequently reducing the upper bound.*

