# OpenReview forum: "From Mismatch to Harmony: Resolving Feature-Classifier Mismatch in Federated Learning via Prompt-Driven Feature Transformation"
_ICLR.cc/2025/Conference — ICLR 2025 Conference Withdrawn Submission_

### Official Review · Reviewer_zNxQ · 2024-10-29

**Soundness:** 2
**Presentation:** 2
**Contribution:** 2
**Rating:** 3
**Confidence:** 4

**Summary:**

The paper proposes to tackle the heterogeneity in federated learning (FL) by "aligning" the features. This is motivated from their experiments, where they noticed that a FedAvg trained model has a better feature extractor than a personalized FL baselines module some linear transformation per client. They learn this feature transformation by conditioning it on a "prompt", which is just a client-specific parameter. The authors also augment the loss with a contrastive loss that uses another prompt. This method is shown to perform well on non-iid versions of CIFAR-10/100 and Tiny ImageNet.

**Strengths:**

- The method proposed is simple yet effective, as evidenced from the accuracy gains.
- There are various experiments, and FedPFT consistently ranks at the top.
- Code is provided, which is great.
- Capturing the heterogeneity of the client via a personalized prompt and a prompt-conditioned feature transformation is a sound approach.

**Weaknesses:**

- The writing is not a strength of the paper. The methodology section is verbose/redundant, especially the problem formulations. For example, authors can benefit from bringing some material from the appendix instead, such as Figure 6, since this comparison was used extensibly in the tables. Some of the claims made by the authors are not scientific, such as "... enhances the synergy between the feature extractor and classifier", which was mentioned multiple times but not precisely defined. It was also not clear what the prompt is until I saw the code. The term "prompt" is not well-defined in this setting, and assumes the reader is familiar with literature involving prompts (i.e., LLMs). Additionally, the authors write the algorithm twice (once right before Sec 3.2. and another in Algorithm 1). This makes the steps shown before Sec 3.2. unnecessary. The related work section is oddly split into two small equally-sized sections, where the most relevant related work (prompt-based) is put in the appendix. Overall, this paper would greatly benefit from an improved writing.
- The authors would also greatly benefit from expanding on the toy problem shown in Figure 1. Could the authors construct a synthetic dataset that reproduces this misalignment? In this case, the authors can provide actual visualizations of allevaiation of this misalignment by FedPFT vs. other personalized baselines where this misalignment occurs. This would be a convincing proof of concept.
- The prompt increases the model's size (specifically, the classifier's size and also adds an additional state to the client). A simple baseline to counter this argument is fine-tuning a local adaptor that has the same number of parameters introduced from using a prompt. An additive adaptor would be sufficient for a fair comparison. The authors are encouraged to check with this simple baseline.
- I don't see the implementation of the other personalized baselines in your code nor the scripts for replicating the experiments.
- Not sure what the contribution behind the addition of the contrastive loss is. This can be done for other methods as well, so it is not an inherent benefit to FedPFT. Table 4 shows that its boost is not as significant as the prompt's.
- The t-SNE visualizations are not conclusive, but they occupy a good portion of the paper. The important metric for comparison is the test accuracy, which is already provided in the tables. I believe putting this in the Appendix is better, and also providing the visualizations for the other competitive baselines (such as FedRoD and pFedSD) would be nice.
- Not sure about the benefits of adding appendix P. The convergence bound is generic and not informative and the discussed setup. No FedPFT-specific step or parameter shows up in the analysis nor the convergence bound.  Also,. the remark following the bound treats R as a problem-specific parameter, but R is a free positive variable appearing from Peter-Paul's inequality and is not problem-specific.

**Questions:**

- The prompt can be used in other ways. For example, it can be used to condition all the layers, condition batch norm layers, etc. Have you tried other approaches in your experiments? Were there any approaches that didn't work? I'm asking because what doesn't work often does not get reported, which makes other researchers repeat the same mistakes.
- Another baseline that makes sense is a classifier personalized with a proximal-style algorithm such as Ditto. In other words, you locally train the classifier and regularize it so that it is close to the FedAvg classifier. It would be great if the authors tried this baseline, or at least provide a reason why it is unnecessary.

---

### Official Review · Reviewer_9Sf4 · 2024-11-03

**Soundness:** 4
**Presentation:** 4
**Contribution:** 4
**Rating:** 8
**Confidence:** 4

**Summary:**

The paper studies personalized federated learning to improve client accuracy. The basic idea is to align the features from the feature extractor and the classifier for each client using a prompt-based approach. The proposed method FedPFT fed personalized prompts into a shared self-attention-based module, where features are transformed via the attention mechanism to align with the global classifier.
Experiments on CIFAR100 and TinyImagenet demonstrate that FedPFT can outperform state-of-the-art parameter-decoupling based approach.

**Strengths:**

- an interesting observation that features from FedAvg are misaligned with the global classifier
- the designed prompt-based module shows promising empirical results
- the ablation study supports the claim that the prompt-based module helps the feature extractor to compute better features that are easier to separate.

**Weaknesses:**

Overall, I think the design of the paper is well-motivated, and the empirical study also show evidence on why the proposed approach can improve the personalized accuracy of clients. With that said, there are still several critical experiments missing to make the paper more convincing.

- Due to the abundant literature on personalized FL and in particular, the parameter decoupling and finetuning-based work, it is not clear to me the critical difference between the proposed approach and existing works that make the proposed prompt-driven approach consistently better. Below are the details.

From the parameter-efficient finetuning literature, there seems to be an understanding that the three categories of approaches
(1) finetuning input prompts (prompt finetuning),
(2) finetuning adaptors attached to the original model, or
(3) finetuning last few layers/initial few layers/or BN layers of a model
can achieve the similar objective of adapting a pre-trained model to the domain-specific task.
Which methods win depends on the task/dataset at hand.
From this perspective, the proposed approach and other finetuning work mentioned in the Appendix (SGPT, FedOTP, FedAPT, pFedPT) belong to (1), works that are compared in the eval such as FedPer, FedBN, etc belongs to (3).
The FedAvg+Linear variant studied in Appendix F belongs to (2).

Both the proposed approach and the counterparts for comparison are variants of (1)(2)(3) that address the feature misalignment problem such that the finetuned model adapts to the client's distribution better. Since it is difficult to judge why one approach in (1)(2)(3) categories will dominate the others, the rationale why the proposed approach can consistently outperform baselines in Table 2 needs more justification.
Specifically, the authors mention "these methods do not completely resolve the mismatch issue during training". The question is why proposed approach address the mismatch issue? To understand the question, I want to make an analogy. Why adaptor-based approach in (2) or the finetuning-based approach in (3) cannot address the same problem addressed by the prompt-based approach in (1)?


- Can the author compare the proposed approach with prompt-based methods enumerated in Appendix B? from the high-level, I cannot tell the difference between using the prompt-based approach for feature alignment (this work) and for finetuning a model (the listed work in appendix B). Why cannot these approach be adapted to the FL setting for personalization -- which can be treated as finetuning the globally-trained model?

- Another baseline to compare with is to directly finetune the classification layer for personalization AFTER the FL training rounds finish. Note that this is different from finetuning the classification layer at each FL round. If the feature extracted from the FedAVG trained global model is indeed good for classification, then the train-then-personalize would be a strong baseline too. Can the author compare with the approach that finetunes the classification layer during training and then personalize the classification layer after training ends?

Below are some minor concerns that may or may not need to be addressed:
- The experiments could use a scalability study. The number of clients used in the work is 40 clients, which can be considered small for cross-device federated learning. Evaluations that demonstrate that the proposed approach can outperform baselines for more clients (e.g., 500, 1000) can be helpful.

- Besides, how about the participation rates? Does the approach assume all the clients will participate in the training? This assumption won't be true in cross-device FL either. This question is relevant since each client needs to store its prompt. If a client didn't participate in the training every FL round, then the prompt will be stale and potentially affect the feature alignment training phase.

- A more intuitive way to show overfitting is to show the train/test accuracy convergence curve -- that is how the generalized accuracy/personalized accuracy on train/test data split change with more FL rounds. I don't see any of these curves in the main paper or the appendix. Can the author provide the train/test accuracy convergence curves across FL rounds?

**Questions:**

see above.

---

### Official Review · Reviewer_Bxvn · 2024-11-03

**Soundness:** 2
**Presentation:** 3
**Contribution:** 2
**Rating:** 3
**Confidence:** 4

**Summary:**

This paper introduces FedPFT, which leverages personalized prompts to resolve the mismatch problem. These prompts, along with local features, are fed into a shared self-attention-based module, where features are transformed via the attention mechanism to align with the global classifier.

**Strengths:**

- This paper is generally well written.
- The proposed method seems to have good results in the experiments.

**Weaknesses:**

- **Statement about pFL**. This paper may have an inappropriate statement about personalized federated learning (pFL). It was said that due to the global model's degraded performance, pFL was proposed. Actually, pFL is not the consequence of poor generalization. Generalization and personalization are two parallel and equally important objectives in FL. FedRoD [1] is recommended for the authors to have a better understanding about the relationship between generalization and personalization.
- **Novelty**.
   - First, the phenomena which the authors claim to find are not new. FedAvg is always to be as a strong baseline if the global model is locally fine-tuned. The authors also can refer to FedRoD [1], so it is not surprising that the authors find the feature extractor of FedAvg is strong enough to have a further personalization. Also, the questions of "classifier biases" and "Feature-Classifier Mismatch" have been studied in many previous works, but the authors lack sufficient discussion and comparison. The question is first proposed in CCVR [2], and later, methods like FedETF [3] and FedFA [4] are proposed.
   -  FedFA uses feature anchors to solve this problem. Especially in FedETF, the projection layer is proposed to map the feature extractor to the classifier, which is similar to FTM in this paper. But unfortunately, neither FedETF nor FedFA are not compared in the experiments.
- **Lack of essential baselines**. For pFL under class imbalance, some essential baselines are missing, like [5, 6].
- **Minor: citation format**. The authors should correctly use _"\citep"_ and _"\citet"_ in the ICLR format. Most of the citations need to be used in _"\citep"_ if they are not used as subjects or objects.

---
[1] Chen, Hong-You, and Wei-Lun Chao. "On Bridging Generic and Personalized Federated Learning for Image Classification." International Conference on Learning Representations 2022.

[2] Luo, Mi, et al. "No fear of heterogeneity: Classifier calibration for federated learning with non-iid data." Advances in Neural Information Processing Systems 34 (2021): 5972-5984.

[3] Li, Zexi, et al. "No fear of classifier biases: Neural collapse inspired federated learning with synthetic and fixed classifier." Proceedings of the IEEE/CVF International Conference on Computer Vision. 2023.

[4] Zhou, Tailin, Jun Zhang, and Danny HK Tsang. "FedFA: Federated learning with feature anchors to align features and classifiers for heterogeneous data." IEEE Transactions on Mobile Computing (2023).

[5] Zhang, Jianqing, et al. "Gpfl: Simultaneously learning global and personalized feature information for personalized federated learning." Proceedings of the IEEE/CVF International Conference on Computer Vision. 2023.

[6] Zhang, Jianqing, et al. "Fedala: Adaptive local aggregation for personalized federated learning." Proceedings of the AAAI Conference on Artificial Intelligence. Vol. 37. No. 9. 2023.

**Questions:**

See weaknesses.

---

### Official Review · Reviewer_4EXC · 2024-11-04

**Soundness:** 3
**Presentation:** 3
**Contribution:** 2
**Rating:** 3
**Confidence:** 3

**Summary:**

The paper describes a new method called FedPFT that is aimed to further improve federated learning, with a motivating example discusses about the mismatch of the features. The method is reasonably aligned with the motivation, and the empirical results demonstrate the strong performances. Most parts of the writing is easy to follow.

**Strengths:**

1. the problem is built upon an important problem, the mismatch of the features. Intuitively, the problem does exist in FL settings, and believed to be an important challenges for FL.

2. The figure illustration makes the paper fairly easy to follow, especially Figure 1, Figure 2, and Figure 6 in appendix.

**Weaknesses:**

1. The motivation seems to be detached from the proposed methods. While the motivation (illustrated in Figure 1) is an interesting problem, the paper does not seem to answer that why the proposed method is THE solution of this problem. The authors demonstrated that the proposed method turns out to be able to solve this problem, but the introduction of the method design can be benefited more from writing closely connected to the motivating examples. I will recommend the authors to offer more detailed discussions to connect these.

2. Similarly, while the results in Figure 4 shows quite well that the proposed method can solve this motivated problem, it does not answer the question that why all the components are necessary to solve this problem. I would recommend the authors to show Figure 4 visualizations across all the settings across Table 4, highlighting the close connections between the motivating of the problem to the solution.

3. The paper has some theoretical discussions in the appendix, but that does not really have much to do with how the proposed method will solve the motivating problem, but about the convergence of the proposed method, seemingly an extension of fairly standard FL convergence proof.

4. The texts involving the contrastive learning part seems to distracting the main message, I feel the paper might deliver the method more clearly without this part. If the authors believe this part is necessary, some additional explanations can help.

**Questions:**

1. The background of prompt is probably necessary for this paper to be self-complete. Thus, I recommend the authors to add some more background sections regarding this.

2. Without sufficient background information of how the prompt is used, and the importance of it, it's hard to appreciate the novelty of this paper. At this moment, I have the question about contrasting the proposed method with many FL literature that uses adaptor-style, partial parameters of the model, to achieve personalized FL. The authors might need to empirically compare to these methods or offer some explanations why the authors believe such comparisons are not necessary.

3. This also leads to the question that the papers seems to compare to an underwhelming set of FL literature, recent personalized FL literature are not discussed or compared. Similarly, the authors might need offer more discussions why they believe such comparisons are not necessary.

**Details Of Ethics Concerns:**

None noted.

---

### Official Review · Reviewer_BeE1 · 2024-11-06

**Soundness:** 1
**Presentation:** 2
**Contribution:** 1
**Rating:** 1
**Confidence:** 4

**Summary:**

This paper proposes a method to improve personalized federated learning (PFL) models by addressing the mismatch between local features and the classifier in the global model. Leveraging the stronger feature representation of a global model trained via federated learning (FL), the authors introduce a prompt-based, two-step local update process. Their approach, FedPFT, uses personalized prompts fed into a self-attention-based module, aligning local features with the global classifier to enhance performance across clients. This method aims to reduce overfitting and improve task flexibility, even in heterogeneous data scenarios

**Strengths:**

To my knowledge, no prior work in the FL area has proposed a prompt-based methodology. If this is accurate, the proposed approach is novel in its methodological perspective within federated learning.

**Weaknesses:**

# Presentation

**The Abstract and Introduction inaccurately describe GFL and PFL**

***(L15-17, L43-48).***
GFL and PFL are distinct fields within FL, each with unique objectives: PFL aims to create personalized models with high accuracy on individual clients' test data, while GFL seeks to build a central model that performs well on a target test dataset. Despite both tasks being decentralized, they share the purpose of leveraging a central model to aggregate client knowledge, producing better feature representations or classifiers than small, local models. **Contrary to the paper’s claim, PFL did not simply emerge to address performance issues from data heterogeneity of GFL; rather, it was developed with these specific, differentiated goals.**

***(L18-19, L53-70).*** The claim that FedAvg’s feature extractor outperforms many PFL methods is not surprising, as this was already shown in the ICLR 2022 FedBABU [1] . In addition to demonstrating the performance of the FedBABU model, the paper also noted that a fine-tuned FedAvg model performed better than previous one-step PFL methods. FedBABU proposed a 2-step approach: training a global model with a shared, frozen classifier across clients to enhance feature alignment under data heterogeneity, followed by local fine-tuning for personalization. Given these similar findings in FedBABU, which are documented in Table 4 and Table 5, the novelty of these results in this paper should be reconsidered.  **For these reasons, the authors should consider adding a row for FedBABU in Table 1 and tempering their claims regarding the novelty of their findings.**

***Table 1***

The difference between probe accuracy and match accuracy is minimal, which diminishes the purpose of the matching approach. Additionally, fine-tuning both the feature extractor and classifier, as suggested in FedBABU, would likely yield better results than updating only the classifier with linear probing. **A row for fine-tuning should also be added to Table 1.**

***Overview Figure 1***

**The overview figure lacks emphasis on the main strength of this paper: the novel introduction of prompts in an FL context.** Instead, it highlights the less surprising finding of feature mismatching. Additionally, it is unclear from the figure whether the matched approach (b) performs better than the mismatched approach (a), as both show 100% accuracy. **Although the intent seems to be to emphasize improved inter-class similarity, the figure does not convey an advantage in terms of the final objective of accuracy, as both cases achieve perfect scores.**

***(L292-293)***

The paper highlights the flexibility of FedPFT in integrating with contrastive learning, with emphasis on this in the main table, introduction, abstract, and analysis sections. **However, there is no detailed explanation of how the contrastive loss is applied in the main text; it is only briefly covered in the appendix, where the explanation is also unclear.** Please provide a clearer, more detailed description in the main text on how the proposed contrastive loss is calculated. **Additionally, prior work in FL with contrastive learning, such as MOON [2], should also be referenced.**


***Table 2***

**The paper reports PFL model performance but includes only FedAvg, a GFL model.** As in Table 5 of [2], please include the performance of PFL models obtained by fine-tuning GFL models, such as FedAvg-FT and FedBABU-FT. At a minimum, I suggest including these two baselines, FedAvg-FT and FedBABU-FT, to provide a more comprehensive comparison.


# Insufficient Related Work
**Related Work Section**

The Related Work section should be divided into two subsections: Global Federated Learning(GFL) and Personalized Federated Learning(PFL).

**In the GFL subsection, include studies on feature enhancement and classifier freezing approaches in FL**. Relevant papers include FedDecorr [3] and FedFN [4], which address feature mismatch issues: FedDecorr examines feature dimensional collapse under data heterogeneity, while FedFN highlights feature norm discrepancies between observed and missing classes in local models. Additionally, include works that explore classifier freezing in FL, such as FedBABU [1], which introduced a shared, frozen classifier across clients to improve feature-classifier alignment. Building on this approach, SphereFed [5], FedETF [6], and FedDr+ [7] propose loss functions or model modifications to further enhance feature-classifier alignment.

**In the PFL subsection,  mention parameter decoupling methods and the 2-step approach of freezing the classifier to create a global model, followed by fine-tuning for personalization, as proposed in [1,4,5,6,7].**

[1] FedBABU: Toward Enhanced Representation for Federated Image Classification, ICLR 2022

[2] Model-Contrastive Federated Learning, CVPR 2021

[3] Towards Understanding and Mitigating Dimensional Collapse in Heterogeneous Federated Learning, ICLR 2023

[4] FedFN: Feature Normalization for Alleviating Data Heterogeneity Problem in Federated Learning. NeurIPS Workshop 2023, Federated Learning in the Age of Foundation Models.

[5] SphereFed: Hyperspherical Federated Learning, ECCV 2022

[6] No Fear of Classifier Biases: Neural Collapse Inspired Federated Learning with Synthetic and Fixed Classifier, ICCV 2023

[7] FedDr+: Stabilizing Dot-regression with Global Feature Distillation for Federated Learning. FedKDD 2024

**Questions:**

While I currently have several concerns that have led to a lower score, I am open to increasing the score if these issues and weakness are adequately addressed during the rebuttal period.

- **(L230)** What is the dimension of $h_k $? It seems to be $(1 + n_k)m \times C $, which would make it significantly larger than the original classifier dimension $ m \times C $, resulting in inefficiency in terms of model parameters. Is this correct?

- **(Prompt Number Effect)** The number of prompts for client k  is defined as $n_k$ (L218). Are these prompts differentiated across clients? Additionally, since $n_k$ appears to be a hyperparameter, how should it be chosen?

- **(L230, Two-step Local Update)** When building the PFL model, the local update process is structured in two steps, similar to [1], with local epochs for each step defined as $ R_f$ and $R_a $. While previous methods used a single hyperparameter $R = R_f + R_a$, this approach introduces two. I feel this may reduce efficiency. Could you provide guidelines on how to allocate $R_f $ and $ R_a $ given an arbitrary \( R \) value?

- **(Dot-Regression vs Prompt Approach)** Although incorporating prompts appears novel, prompts add input and training costs. The main goal, as shown in Figure 1, seems to be increasing inter-class diversity for features within local clients. In [2], Dot-Regression [3] is proposed for this purpose by fine-tuning the global model beacause it is optimize for inter-class diversity perspective purpose.. Does the prompt-based approach offer a clear advantage over this philosophy?

[1] No Fear of Classifier Biases: Neural Collapse Inspired Federated Learning with Synthetic and Fixed Classifier, ICCV 2023

[2] FedDr+: Stabilizing Dot-regression with Global Feature Distillation for Federated Learning. FedKDD 2024

[3] Inducing Neural Collapse in Imbalanced Learning: Do We Really Need a Learnable Classifier at the End of Deep Neural Network?, NeurIPS 2022

---

### Note · Authors · 2024-11-15

**Comment:**

We sincerely thank all the reviewers for dedicating their valuable time to evaluate our paper. We will carefully consider the reviewers’ insightful comments to improve our manuscript.

**Withdrawal Confirmation:**

I have read and agree with the venue's withdrawal policy on behalf of myself and my co-authors.